# Strategyproof Voting under Correlated Beliefs

**Daniel Halpern**
Harvard University
dhalpern@g.harvard.edu

**Rachel Li**
Harvard University
rachelli@college.harvard.edu

**Ariel D. Procaccia**
Harvard University
arielpro@seas.harvard.edu

## Abstract

In voting theory, when voters have ranked preferences over candidates, the celebrated *Gibbard-Satterthwaite Theorem* essentially rules out the existence of reasonable strategyproof methods for picking a winner. What if we weaken strategyproofness to only hold for Bayesian voters with beliefs over others' preferences? When voters believe other participants' rankings are drawn independently from a fixed distribution, the impossibility persists. However, it is quite reasonable for a voter to believe that other votes are correlated, either to each other or to their own ranking. We consider such beliefs induced by classic probabilistic models in social choice such as the *Mallows*, *Placket-Luce*, and *Thurstone-Mosteller* models. We single out the plurality rule (choosing the candidate ranked first most often) as a particularly promising choice as it is strategyproof for a large class of beliefs containing the specific ones we introduce. Further, we show that plurality is unique among positional scoring rules in having this property: no other scoring rule is strategyproof for beliefs induced by the Mallows model when there are a sufficient number of voters. Finally, we give examples of prominent non-scoring voting rules failing to be strategyproof on beliefs in this class, further bolstering the case for plurality.

## 1 Introduction

One of the most celebrated results in voting theory is the *Gibbard-Satterthwaite Theorem* [8, 22]. It states that when voters express ordinal preferences over at least 3 candidates, there is no "reasonable" aggregation rule that is *strategy-proof*: there will always exist instances where voters will be incentivized to manipulate and lie about their preferences to achieve a better outcome.

However, one caveat about this strong negative result is that, a priori, a voter may need perfect information about how others vote to manipulate successfully. Perhaps, if the voter is slightly uncertain, no manipulation helps consistently enough to be worthwhile. Majumdar and Sen [13] analyzed exactly this question when voters have *independent beliefs*. That is, when a voter is considering whether or not to manipulate, they assume all others have rankings drawn independently from a fixed distribution. The classic notion of strategyproofness no longer makes sense in this probabilistic Bayesian setting, so they instead use the natural extension known as *ordinally Bayesian incentive compatible (OBIC)*, essentially that the rules are strategyproof in expectation no matter what underlying cardinal values voters have. Their results, unfortunately, are widely negative. They show that for a "large" set of distributions, Gibbard-Satterthwaite still holds. There do exist distributions where many rules are OBIC, e.g., the uniform distribution over all rankings. Still, these positive examples are extremely brittle: even a slight perturbation leads back to the impossibility.

37th Conference on Neural Information Processing Systems (NeurIPS 2023).

But independent beliefs are quite restrictive. They cannot capture several kinds of beliefs that would likely occur in practice. For one, when the number of voters is large, the uncertainty essentially vanishes. Suppose a distribution places probability $1/4$ on other voters having the ranking $a \succ b \succ c$. In that case, when the number of voters is large, it is extremely unlikely that the proportion of voters with this ranking is anything other than $1/4 \pm \varepsilon$. In a real presidential election, a voter may quite plausibly believe that a candidate will receive anywhere between 45% and 55% of the votes, but this situation simply cannot be captured by a single independent ranking distribution. Second, one's own ranking may influence the probability placed on others. Suppose a voter, after much research, discovers that they prefer one proposal to another; they may reasonably believe others are clever enough to have reached a similar conclusion. In terms of their beliefs, they may place a slightly higher probability on others voting more similarly to them than not, no matter what their realized preferences are.

This has led follow-up work to consider the same question under *correlated beliefs* [1, 16, 14, 2]. However, besides some impossibilities, the work so far has largely been of the following form: for any reasonable voting rule, *there exists* a set of beliefs where the rule is OBIC. But perhaps the more natural direction is the converse: under a natural set of beliefs, is there a reasonable voting rule that is consistently OBIC? Can this property help us distinguish between voting rules, showing that under some reasonable beliefs, certain rules are *not* OBIC, thereby bolstering the case for the provably incentive-compatible ones? These are the questions we tackle.

**Our contributions.** We begin by presenting various classes of beliefs induced by classic probabilistic social choice models such as the *Mallows* [15], *Thurstone-Mosteller* [23, 18], and *Placket-Luce* [20, 12] models. In essence, these are the beliefs a voter would have if they assume that voter preferences were generated by such a model. Inspired by these models, we present a novel class of mildly correlated beliefs that includes all of them. We show that, under this class of beliefs, the *plurality* rule is OBIC.

Next, we provide a negative result: Among positional scoring rules (where each voter assigns a fixed score to each position in their ranking), plurality is unique in being OBIC when voters have Mallows beliefs. All other rules will become not OBIC when there are three candidates, at least when there are a sufficient number of voters. In addition, we provide some robustness checks on this negative result. A popular positional scoring rule known as *Borda Count* fails for any number of voters. By contrast, we identify other positional scoring rules that are OBIC with two voters, meaning our result could not be strengthened by relaxing the sufficient number of voters requirement.

Finally, we complement this more sweeping classification with examples of other prominent rules, such as *Copeland* and *maximin*, which fail to be OBIC with specific Mallows beliefs and few voters. This further bolsters the case for plurality as an unusually attractive rule when viewed through the lens of ordinal Bayesian incentive compatibility under correlated beliefs.

**Related work.** As mentioned above, the analysis of OBIC voting rules began with Majumdar and Sen [13] essentially providing the final word on independent beliefs; their notion of OBIC dates back to work on committee selection [5].

Since then, there have been a few lines of work on correlated beliefs with slightly different goals. The most closely related is that of Majumdar and Sen [14]. They define a large class of positively correlated beliefs based on the Kemeny metric and then show in a similar fashion to the Gibbard-Satterthwaite Theorem that any voting rule that is OBIC with respect to these beliefs, along with being Pareto efficient, is necessarily dictatorial. They do present one voting rule that is both OBIC with respect to these beliefs and nondictatorial (while not being Pareto efficient), but it is a clearly impractical rule that is designed to make a technical point.[1] Note that all the rules we consider are Pareto efficient.

Another line of work considers *local* OBIC. A voting rule is locally OBIC with respect to a class of beliefs if there *exists* a belief in the class such that any belief in a neighborhood of the original is OBIC. This means the rule remains OBIC even after a slight perturbation to the underlying belief. Bhargava et al. [1] and Bose and Roy [2] attempt to classify the set of locally OBIC voting rules with

---

[1]Their rule is called *Unanimity with Status Quo* . There is one default candidate $x$. If there is a candidate $y$ which every single voter places as their top choice, then $y$ is elected, but in any other case, $x$ wins.

respect to a large class of correlated beliefs and show that under minimal conditions, this requirement can be satisfied.

Mandal and Parkes [16] consider a different notion of incentive compatibility which, rather than requiring that no manipulation can lead to a utility gain in expectation, bounds the probability under which there is a utility gain. They again do this with respect to several different classes of beliefs, including one that we consider based on the Mallows Model.

Further afield, there are extensive lines of research on circumventing the Gibbard-Satterthwaite Theorem. We provide examples of three here, although there are many others. One line considers the complexity, showing that some rules, while in principle susceptible to manipulation, have instances where it is hard (in the worst case) to find such a manipulation [7, 3]. Another considers strategyproofness under restricted domains, where a voter's set of possible rankings is limited [7, 3]. A third considers the likelihood of an individual arriving in an instance where they are able to manipulate at all [17, 25].

Finally, without considering strategyproofness, there has been much work on probabilistic social choice, making use of the models on which our results are based, especially in learning preferences from data [11, 24, 10, 19].

## 2 Model

We begin by introducing the classic social model and then later describe relevant definitions for social choice under uncertainty.

**Classic social choice model.** Let $N = \{1, \ldots, n\}$ be a set of $n$ *voters*, and let $\mathcal{A} = \{a_1, \ldots, a_m\}$ be a set of $m$ *alternatives*. Let $\mathcal{L}$ be the set of rankings over $\mathcal{A}$, where for $\sigma \in \mathcal{L}$, $\sigma(j)$ is the $j$'th candidate in ranking $\sigma$ and $\sigma^{-1}(a)$ is the ranking index of candidate $a$. We use the notation $a \succ_\sigma b$ to denote that $\sigma^{-1}(a) < \sigma^{-1}(b)$ and $a \succeq_\sigma b$ to denote $\sigma^{-1}(a) \leq \sigma^{-1}(b)$, i.e., $a$ is strictly (or weakly) preferred to $b$ under $\sigma$. Additionally, instead of writing $\sigma = a \succ b \succ c$, when it is clear from context, we will sometimes shorten this to $\sigma = abc$. Each voter $i$ has a ranking $\sigma_i \in \mathcal{L}$ and the tuple of these rankings $\boldsymbol{\sigma} = (\sigma_1, \ldots, \sigma_n) \in \mathcal{L}^n$ is called the *preference profile* . We let $\boldsymbol{\sigma}_{-i} \in \mathcal{L}^{n-1}$ denote the profile without voter $i$, and for a ranking $\sigma_i' \in \mathcal{L}$, we let $(\boldsymbol{\sigma}_{-i}, \sigma_i')$ be the profile with $\sigma_i$ replaced with $\sigma_i'$.

A *voting rule* is a function $f$ that, given a profile $\boldsymbol{\sigma}$, outputs a distribution over winning alternatives. We define several voting rules of interest here. Our theoretical results will primarily focus on *positional scoring rules* [26]. A positional scoring rule $f$ is parameterized by a vector of $(s_1, \ldots, s_m)$ where each $s_j \in \mathbb{Z}_{\geq 0}$ with $s_1 \geq \cdots \geq s_m$ and $s_1 > s_m$. On a profile $\boldsymbol{\sigma}$, for each voter $i$, their $j$'th candidate $\sigma_i(j)$ is given $s_j$ points. The points are added up over all voters, and the winning candidate is the one with the most points. More formally, for a ranking $\sigma \in \mathcal{L}$ and candidate $c \in \mathcal{A}$, we write $\text{sc}_c^f(\sigma) = s_{\sigma^{-1}(c)}$ for the points (or score) given to $c$ by $\sigma$. For a profile $\boldsymbol{\sigma}$, we write $\text{sc}_c^f(\boldsymbol{\sigma}) = \sum_i \text{sc}_c^f(\sigma_i)$ to be the total points. When $f$ is clear from context, we may drop it from the notation. In deterministic settings, when there is a tie, a tie-breaking rule needs to be given (i.e., tie-break in favor of lower index candidates). Since we will be working in a probabilistic setting, it will be more convenient to assume *uniform random tie-breaking*, so that if there is a tie among $k$ candidates, each wins with probability $1/k$. However, our results would continue to hold even with arbitrary deterministic choices. Two rules of particular interest are *plurality*, parameterized by the vector $(1, 0, \ldots, 0)$, and *Borda count*, parameterized by the vector $(m - 1, m - 2, \ldots, 1, 0)$.

We consider two additional rules beyond positional scoring rules, *Copeland* and *maximin*. To define them, for a profile $\boldsymbol{\sigma}$ we define the pairwise margin for two candidates $a$ and $b$, $N_{ab}(\boldsymbol{\sigma}) = |\{i | a \succ_i b\}|$, i.e., the number of voters that prefer candidate $a$ to candidate $b$.

For Copeland, we define the Copeland score for a candidate $a$ as $\sum_{b \neq a} \mathbf{I}[N_{ab}(\boldsymbol{\sigma}) > n/2] + (1/2)\mathbf{I}[N_{ab}(\boldsymbol{\sigma}) = n/2]$. In words, the candidate gets one point for every other candidate they pairwise beat and a half point for every other candidate they pairwise tie. The Copeland winners are those with the highest Copeland scores (with uniform tie-breaking). For maximin, we define the maximin score for a candidate $a$ as $\min_{b \neq a} N_{ab}(\boldsymbol{\sigma})$, i.e., the smallest margin by which $a$ beats another candidate. Again, the maximin winners are those with the highest maximin scores (with uniform tie-breaking).

A voting rule $f$ is called *strategy-proof* if for all profiles $\boldsymbol{\sigma}$, all voters $i$, and all alternative manipulations $\sigma_i' \in \mathcal{L}$, $f(\boldsymbol{\sigma}) \succeq_{\sigma_i} f(\boldsymbol{\sigma}_{-i}, \sigma_i')$. That is, no voter can ever improve the outcome of the vote by misreporting. A rule is called onto if for all candidates $a \in \mathcal{A}$, there is a profile $\boldsymbol{\sigma}$ where $f(\boldsymbol{\sigma}) = a$, and is called dictatorial if there is a voter $i$ for which $f(\boldsymbol{\sigma}) = \sigma_i(1)$, i.e., voter $i$ always gets their top choice. The Gibbard-Satterthwaite Theorem states that any rule for $m \geq 3$ candidates that is strategy-proof and onto is necessarily dictatorial. Since "reasonable" rules must be onto and nondictatorial, this eliminates the possibility of any being strategy-proof.

These notions can also be extended to randomized rules. A *utility function* $u$ is a mapping from candidates to real numbers. We say that $u$ is consistent with a ranking $\sigma$ if $u(x) > u(y) \Leftrightarrow x \succ_{\sigma} y$. A (randomized) voting rule $f$ is called SD-strategy-proof if for all profiles $\boldsymbol{\sigma}$, all voters $i$, all manipulations $\sigma_i'$, and all utility functions $u$ consistent with $\sigma_i$, $\mathbb{E}[u(f(\boldsymbol{\sigma}))] \geq \mathbb{E}[u(f(\boldsymbol{\sigma}_{-i}, \sigma_i'))]$. This says that no matter what underlying utilities an agent has, as long as they are consistent with their ranking, they cannot improve their expected utility by manipulation. An equivalent definition can be given with respect to stochastic dominance (hence the SD in the name). For $k \leq m$, let $B_k(\sigma) = \{\sigma(1), \ldots, \sigma(k)\}$ be the set of the $k$ best alternatives according to $\sigma$. Then, SD-strategy-proofness can be rephrased as requiring that for all profiles $\boldsymbol{\sigma}$, all voters $i$, all manipulations $\sigma_i'$, and all $k \leq m$, $\Pr[f(\boldsymbol{\sigma}) \in B_k(\sigma_i)] \geq \Pr[f(\boldsymbol{\sigma}_{-i}, \sigma_i') \in B_k(\sigma_i)]$.

We say a rule $f$ is unilateral if it depends only on a single voter, i.e., there is a voter $i$ such that for all $\boldsymbol{\sigma}$ and $\boldsymbol{\sigma}'$, if $\sigma_i = \sigma_i'$, then $f(\boldsymbol{\sigma}) = f(\boldsymbol{\sigma}')$. A rule $f$ is called a duple if its range is two candidates, i.e., there is a pair of candidates $a$ and $b$ such that for all $\boldsymbol{\sigma}$, $f(\boldsymbol{\sigma}) \in \{a, b\}$. Gibbard [9] extended the Gibbard-Satterthwaite theorem to randomized rules as follows: Any randomized rule $f$ that is strategyproof is a mixture over unilateral and duple rules. Since unilateral and duple rules are seen as undesirable, this implies that finding a reasonable, strategy-proof voting rule, even allowing randomization, is a hopeless endeavor.

**Social choice under uncertainty.** A *belief* for voter $i$ is a probability measure $\mathbb{P}_i$ over the set of profiles $\mathcal{L}^n$ (when the $i$ is clear from context we will drop it from the notation). This describes $i$'s prior probability over profiles before considering their own ranking. After observing their own ranking $\hat{\sigma}_i$, the voter can update their posterior using the conditional distribution $\mathbb{P}[\cdot \mid \sigma_i = \hat{\sigma}_i]$. For notational convenience, we will often shorten this to $\mathbb{P}[\cdot \mid \hat{\sigma}_i]$.

In this model, we need a slightly different notion of strategyproofness. A voting rule is called *ordinally Bayesian incentive compatible (OBIC)* with respect to a beliefs $(\mathbb{P}_1, \ldots, \mathbb{P}_n)$ if for all voters $i$, all rankings $\hat{\sigma}_i$, all manipulations $\sigma_i'$, and all utility functions $u$ consistent with $\hat{\sigma}_i$, $\mathbb{E}[u(f(\boldsymbol{\sigma}_{-i}, \hat{\sigma}_i)) \mid \hat{\sigma}_i] \geq \mathbb{E}[u(f(\boldsymbol{\sigma}_{-i}, \sigma_i') \mid \hat{\sigma}_i]$. This is the natural generalization of SD-strategyproofness to a Bayesian setting. Just as with SD-strategyproofness, an equivalent definition is for all voters $i$, all rankings $\hat{\sigma}_i$, all manipulations $\sigma_i'$, and all $k \leq m$, $\mathbb{P}[f(\boldsymbol{\sigma}_{-i}, \hat{\sigma}_i) \in B_k(\hat{\sigma}_i) \mid \hat{\sigma}_i] \geq \mathbb{P}[f(\boldsymbol{\sigma}_{-i}, \sigma_i') \in B_k(\hat{\sigma}_i) \mid \hat{\sigma}_i]$.

We now present a few possible choices of "reasonable" priors based on well-known probabilistic models of social choice. The first is based off of a *Mallows Model* [15]. This model is parameterized by a *ground truth* ranking $\tau \in \mathcal{L}$ and a dispersion quantity $\varphi$. We define the Kendall tau distance between rankings $d(\sigma_1, \sigma_2) = |\{(a, b) \in \mathcal{A}^2 | a \succ_{\sigma_1} b \wedge b \succ_{\sigma_2} a\}|$, i.e., the number of pairs of candidates on which $\sigma_1$ and $\sigma_2$ disagree. In a Mallows Model, each voter's ranking is drawn independently with probability proportional to $\varphi^{d(\sigma, \tau)}$. More formally, the probability that a specific ranking $\sigma$ is drawn is equal to $\frac{\varphi^{d(\sigma, \tau)}}{Z}$ where $Z = \sum_{\sigma \in \mathcal{L}} \varphi^{d(\sigma, \tau)}$ is the normalizing constant. One can easily check that if we extend the notion of Kendall tau distance to operate on a profile and a ranking, with $d(\boldsymbol{\sigma}, \tau) = \sum_i d(\sigma_i, \tau)$, then the probability of sampling a profile $\boldsymbol{\sigma}$ is proportional to $\varphi^{d(\boldsymbol{\sigma}, \tau)}$ (this time with a $Z^n$ normalizing constant).

We convert this model into a prior in two ways. The first we call a *confident Mallows prior* parameterized by $\varphi$. The agent assumes a ground truth $\tau$ is first drawn from some (arbitrary) distribution, then, given this ground truth, $\sigma_i = \tau$ with probability 1 and the remainder of the profile $\boldsymbol{\sigma}_{-i}$ is drawn from a Mallows Model with a fixed $\varphi$ using $\tau$. Essentially, the agent believes that they correctly know the ground truth, but all others only approximate this truth using a Mallows Model. The conditional distribution over the remainder of the profile $\boldsymbol{\sigma}_{-i}$ given $\hat{\sigma}_i$ then follows a standard Mallows model with the ground truth equal to $\hat{\sigma}_i$, so $\mathbb{P}[\boldsymbol{\sigma}_{-i} \mid \hat{\sigma}_i] \propto \varphi^{d(\boldsymbol{\sigma}_{-i}, \hat{\sigma}_i)}$.[2]

---

[2]This is equivalent to the *Conditional Mallows Model* of Mandal and Parkes [16].

The second we call an *unconfident Mallows prior*. Here, the agent believes that the ground truth $\tau$ is drawn uniformly at random, and then the entire profile (including their own ranking) is drawn from a Mallows Model. Therefore, $\mathbb{P}[\boldsymbol{\sigma}] = \frac{1}{m!} \sum_{\tau \in \mathcal{L}} \frac{\varphi^{d_{kt}(\boldsymbol{\sigma},\tau)}}{Z^n}$. Since for any $\hat{\sigma}_i$, by symmetry $\Pr[\sigma_i = \hat{\sigma}_i] = \frac{1}{m!}$, we can write the conditional probability as

$$\mathbb{P}[\boldsymbol{\sigma}_{-i} \mid \hat{\sigma}_i] = \sum_{\tau \in \mathcal{L}} \frac{\varphi^{d((\boldsymbol{\sigma}_{-i},\hat{\sigma}_i),\tau)}}{Z^n} = \sum_{\tau \in \mathcal{L}} \frac{\varphi^{d(\boldsymbol{\sigma}_{-i},\tau)}}{Z^{n-1}} \cdot \frac{\varphi^{d(\hat{\sigma}_i,\tau)}}{Z}.$$

We will abuse notation slightly and write $\mathbb{P}[\tau \mid \hat{\sigma}_i] = \frac{\varphi^{d(\hat{\sigma}_i,\tau)}}{Z}$ and $\mathbb{P}[\boldsymbol{\sigma}_{-i} \mid \tau] = \frac{\varphi^{d(\boldsymbol{\sigma}_{-i},\tau)}}{Z^{n-1}}$, so that

$$\mathbb{P}[\boldsymbol{\sigma}_{-i} \mid \hat{\sigma}_i] = \sum_{\tau \in \mathcal{L}} \mathbb{P}[\boldsymbol{\sigma}_{-i} \mid \tau] \cdot \mathbb{P}[\tau \mid \hat{\sigma}_i].$$

We can interpret this as saying the voter has a posterior over ground truths, $\mathbb{P}[\tau \mid \hat{\sigma}_i]$, and is using this posterior to infer the probability of the rest of the profile. Intuitively, the agent is uncertain over ground truths but, due to the observation of their ranking, places higher weight on ground truths that are closer to their own ranking. This decomposition is possible because the rest of the profile $\boldsymbol{\sigma}_{-i}$ is conditionally independent of $\hat{\sigma}_i$ given the ground truth $\tau$.

The *Thurstone-Mosteller* model is defined with respect to underlying means $\mu_c$ for each candidate $c \in \mathcal{A}$. To sample a ranking, a value $X_c \sim \mathcal{N}(\mu_c, 1)$ is drawn independently for each candidate $c$ from a normal distribution with variance 1 around the mean. The resulting ranking is the order of the $X_c$ values from highest to lowest. The *Placket-Luce* model is defined with respect to underlying weights $w_c > 0$ for each candidate $c \in \mathcal{A}$. To sample a ranking, we iteratively select a candidate $c$ from the remaining unchosen candidates $P$ with probability $\frac{w_c}{\sum_{c' \in P} w_{c'}}$.

To convert these models to beliefs, we assume that the voter believes there are underlying distinct means $\mu_1 > \cdots > \mu_m$ (resp. weights $w_1 > \cdots > w_m$) but is uncertain about which candidate has which mean (resp. weight). To relate this to the Mallows belief, we will call this order $\tau$, the ground truth. In the confident version, the voter believes that their ranking is always equal to $\tau$, but all other votes are drawn from the corresponding model. In the unconfident version, the voter believes a priori that $\tau$ is drawn uniformly at random, and then all voter rankings, including their own, are drawn from the corresponding model. As with the Mallows beliefs, the voter can do a Bayesian update to compute a posterior about which candidate was assigned to which weight. We can again decompose

$$\mathbb{P}[\boldsymbol{\sigma}_{-i} \mid \hat{\sigma}_i] = \sum_{\tau} \mathbb{P}[\boldsymbol{\sigma}_{-i} \mid \tau] \cdot \mathbb{P}[\tau \mid \hat{\sigma}_i],$$

where, by Bayes' rule, $\mathbb{P}[\tau \mid \hat{\sigma}_i] \propto \mathbb{P}[\hat{\sigma}_i | \tau]$, the probability of generating $\hat{\sigma}_i$ under the model with ground truth $\tau$.

Note that to make this more general, it would also make sense for the voter to believe there is a distribution over means or weights; our results continue to hold with this more general class; however, for ease of presentation, we focus on the more restricted form.

## 3 Plurality is OBIC

We start by defining a class of beliefs that we call *top-choice correlated*. The class is similar in spirit (although incomparable) to the class of top-set correlated beliefs introduced by Bhargava et al. [1].

To define the class, given a profile $\boldsymbol{\sigma}$ and a candidate $c \in \mathcal{A}$, we let $\text{PLU}_c(\boldsymbol{\sigma}) = |\{i \mid \sigma_i(1) = c\}|$ be the *plurality score of $c$*, i.e., the number of voters that rank $c$ first. Further, we let $\text{PLU}(\boldsymbol{\sigma})$ be the vector of plurality scores indexed by the candidates. A belief $\mathbb{P}_i$ is top-choice correlated if the following holds. Fix a ranking $\hat{\sigma}_i$ and let $a = \hat{\sigma}_i(1)$. Then, for all candidates $b \neq a$ and all pairs of plurality vectors $\mathbf{r}$ and $\mathbf{r}'$ such that $r_c = r'_c$ for all $c \neq a, b$, $r_a = r'_b$, $r_b = r'_a$, and $r_a > r_b$, $\mathbb{P}[\text{PLU}(\boldsymbol{\sigma}_{-i}) = \mathbf{r} \mid \hat{\sigma}_i] \geq \mathbb{P}[\text{PLU}(\boldsymbol{\sigma}_{-i}) = \mathbf{r}' \mid \hat{\sigma}_i]$. This says that if the voter is told the remaining plurality scores of all other candidates except $a$ and $b$, as well as possible scores for $a$ and $b$, they would think it is more likely that $a$ (their top choice) has the higher score. In other words, all else being equal, the voter's top choice is more likely to perform better than other candidates.

We now claim that all of the specific beliefs we have introduced are top-choice correlated, suggesting that this condition is quite weak.

**Lemma 1.** *The confident and unconfident versions of Mallows, Thurstone-Mosteller, and Placket-Luce beliefs under any parameter settings are top-choice correlated.*

The proof of Lemma 1 can be found in Appendix A. For confident versions of these beliefs, this is relatively straightforward as the models directly place higher mass on the voter's top choice being chosen. For the unconfident versions, slightly more intricate analysis is necessary to show that more mass is placed on ground truth rankings where the voter's top choice is higher, and from this, we can reach the same conclusion.

Despite the breadth of the class of top-choice correlated beliefs, it turns out that plurality is OBIC for all beliefs in this class.

**Lemma 2.** *Under any top-choice correlated beliefs, plurality is OBIC.*

*Proof.* Let $f$ be the plurality voting rule, $i$ be an agent, and $\mathbb{P}$ be their top-choice correlated belief. Suppose $i$ observes $\hat{\sigma}_i$, and let $u$ be an arbitrary utility function that is consistent with $\hat{\sigma}_i$. Let $a = \hat{\sigma}_i(1)$ be their top-ranked alternative.

Let $\sigma_i'$ be a possible manipulation for voter $i$, and let $b = \sigma_i'(1)$ be the top-ranked alternative. Notice that if $a = b$, then the outcome under plurality is identical, and this manipulation cannot be an improvement. Hence, from now on, we assume that $b \neq a$.

For $\boldsymbol{\sigma}_{-i}$, let $\text{UG}(\boldsymbol{\sigma}_{-i}) = \mathbb{E}[u(f(\boldsymbol{\sigma}_{-i}, \sigma_i'))] - \mathbb{E}[u(f(\boldsymbol{\sigma}_{-i}, \hat{\sigma}_i))]$ be the expected utility gain of switching from $\hat{\sigma}_i$ to $\sigma_i'$ when others report $\boldsymbol{\sigma}_{-i}$. We wish to show that $\mathbb{E}[\text{UG}(\boldsymbol{\sigma}_{-i})|\hat{\sigma}_i] \leq 0$, where the expectation is over the belief $\mathbb{P}$. To simplify notation, we will allow the utility function $u$ to operate on (nonempty) sets of candidates $S$ as $u(S) = \frac{1}{|S|} \sum_{c \in S} u(c)$. Note that when the set of plurality winners on a profile is $S$, the expected utility is $u(S)$.

We now partition the possible $\boldsymbol{\sigma}_{-i}$ based on their utility gain. Let $C \subseteq \mathcal{A} \setminus \{a, b\}$ be a (nonempty) set candidates not including $a$ and $b$. For each set $C$, we define eight sets of profiles $\boldsymbol{\sigma}_{-i}$ depending on the winners under $(\boldsymbol{\sigma}_{-i}, \hat{\sigma}_i)$ and $(\boldsymbol{\sigma}_{-i}, \sigma_i')$, $E_1(C)^a, E_1(C)^b, E_2(C)^a, E_2(C)^b, E_3(C)^a, E_3(C)^b, E_4(C), E_5(C)$. In each $E(C)$ set, $C$ will be the set of candidates excluding $a$ and $b$ with the highest plurality score. We abuse notation slightly and write $\text{PLU}(C)$ for the (tied) plurality score of each of these candidates and $\text{PLU}(a)$ and $\text{PLU}(b)$ for the plurality scores of $a$ and $b$, respectively. The sets are otherwise defined by the set of plurality winners in $(\boldsymbol{\sigma}_{-i}, \hat{\sigma}_i)$ and $(\boldsymbol{\sigma}_{-i}, \sigma_i')$. The definitions can be found in Table 1. One can check that these (disjoint) sets collectively cover all possible $\boldsymbol{\sigma}_{-i}$ where $\text{UG}(\boldsymbol{\sigma}_{-i})$ is nonzero.

We can now rewrite the expected utility gain in terms of these sets. For each set $E(C)$, we write $\text{UG}(E(C))$ for the expected utility gain for profiles $\boldsymbol{\sigma}_{-i} \in E(C)$ (which will always be the same for all $\boldsymbol{\sigma}_{-i} \in E(C)$). From this, we have

$$\mathbb{E}[\text{UG}(\boldsymbol{\sigma}_{-i})|\hat{\sigma}_i] = \sum_{\substack{C \subseteq \mathcal{A} \setminus \{a,b\} \\ C \neq \emptyset}} \left( \sum_{j=1}^{3} \left( \mathbb{P}[E_j(C)^a|\hat{\sigma}_i]\text{UG}(E_j(C)^a) + \mathbb{P}[E_j(C)^b|\hat{\sigma}_i]\text{UG}(E_j(C)^b) \right) \right.$$

$$\left. + \mathbb{P}[E_4(C)|\hat{\sigma}_i]\text{UG}(E_4(C)) + \mathbb{P}[E_5(C)|\hat{\sigma}_i]\text{UG}(E_5(C)) \right)$$

Our goal, again, is to show that this expression is at most $0$. Notice that for each $C$, $\text{UG}(E_4(C)) = u(b) - u(a) < 0$ and $\text{UG}(E_5(C)) = \frac{1}{|C|}(u(b) - u(a)) < 0$ because $u(a) > u(c)$ for all other candidates $c$. In what remains, we show that for all $C$ and each $j \leq 3$,

$$\mathbb{P}[E_j(C)^a]\text{UG}(E_j(C)^a) + \mathbb{P}[E_j(C)^b]\text{UG}(E_j(C)^b) \leq 0. \tag{1}$$

Fix an arbitrary $C$. To do this, we show that for each $j$, $\text{UG}(E_j(C)^a) \leq 0$, $-\text{UG}(E_j(C)^a) \geq \text{UG}(E_j(C)^b)$, and $\mathbb{P}[E_j(C)^a] \geq \mathbb{P}[E_j(C)^b]$. Together, these imply (1).

We analyze the case of $j = 1$; the arguments for $j = 2$ and $j = 3$ are very similar. Notice that

$$\text{UG}(E_1(C)^a) = u(C \cup \{a\}) - u(a) = \frac{|C|}{|C| + 1}(u(C) - u(a)),$$

| Set | Condition | $(\boldsymbol{\sigma}_{-i}, \hat{\sigma}_i)$ Winners | $(\boldsymbol{\sigma}_{-i}, \sigma'_i)$ Winners |
|---|---|---|---|
| $E_1(C)^a$ | $\text{PLU}(a) = \text{PLU}(C) > \text{PLU}(b) + 1$ | $a$ | $C \cup \{a\}$ |
| $E_1(C)^b$ | $\text{PLU}(b) = \text{PLU}(C) > \text{PLU}(a) + 1$ | $C \cup \{b\}$ | $b$ |
| $E_2(C)^a$ | $\text{PLU}(a) = \text{PLU}(C) = \text{PLU}(b) + 1$ | $a$ | $C \cup \{a, b\}$ |
| $E_2(C)^b$ | $\text{PLU}(b) = \text{PLU}(C) = \text{PLU}(a) + 1$ | $C \cup \{a, b\}$ | $b$ |
| $E_3(C)^a$ | $\text{PLU}(C) = \text{PLU}(a) + 1 > \text{PLU}(b) + 1$ | $C \cup \{a\}$ | $C$ |
| $E_3(C)^b$ | $\text{PLU}(C) = \text{PLU}(b) + 1 > \text{PLU}(a) + 1$ | $C$ | $C \cup \{b\}$ |
| $E_4(C)$ | $\text{PLU}(a) = \text{PLU}(b) \geq \text{PLU}(C)$ | $a$ | $b$ |
| $E_5(C)$ | $\text{PLU}(C) = \text{PLU}(a) + 1 = \text{PLU}(b) + 1$ | $C \cup \{a\}$ | $C \cup \{b\}$ |

Table 1: Definition of the sets $E_1(C)^a$, $E_1(C)^b$, $E_2(C)^a$, $E_2(C)^b$, $E_3(C)^a$, $E_3(C)^b$, $E_4(C)$, and $E_5(C)$. They contain all $\boldsymbol{\sigma}_{-i}$ that satisfy the corresponding condition. In each set, the contained $\boldsymbol{\sigma}_{-i}$ all have the same winners in both $(\boldsymbol{\sigma}_{-i}, \hat{\sigma}_i)$ and $(\boldsymbol{\sigma}_{-i}, \sigma'_i)$ as seen in the corresponding columns.

and symmetrically

$$\text{UG}(E_1(C)^b) = u(b) - u(C \cup \{b\}) = \frac{|C|}{|C| + 1}(u(b) - u(C)).$$

Since $u(a)$ is maximal, $\text{UG}(E_1(C)^a) \leq 0$. Further,

$$-\text{UG}(E_1(C)^a) - \text{UG}(E_1(C)^b) = \frac{|C|}{|C| + 1}(u(a) - u(b)) \geq 0.$$

Finally, to show $\mathbb{P}[E_1(C)^a] \geq \mathbb{P}[E_1(C)^b]$, we consider the vectors of plurality scores (indexed by candidates) $\mathbf{r}^a$ and $\mathbf{r}^b$ that lead a profile $\boldsymbol{\sigma}_{-i}$ to end in up in $E_1(C)^a$ and $E_1(C)^b$, respectively. Due to the symmetry, there is a natural bijection between these two sets of vectors obtained by swapping the $a$ and $b$ components. Further, by the definition of $E_1(C)^a$ and $E_1(C)^b$, the $a$ component of $\mathbf{r}^a$ is always strictly larger than the $b$ component. Since $\mathbb{P}$ is top-choice correlated, for any two vectors that differ by swapping the $a$ and $b$ components, $\mathbb{P}[\cdot|\hat{\sigma}_i]$ always places higher mass on the vector in $\mathbf{r}^a$. Therefore, $\mathbb{P}[E_1(C)^a|\hat{\sigma}_i] \geq \mathbb{P}[E_1(C)^b|\hat{\sigma}_i]$, as needed. $\square$

From these two lemmas, we immediately derive our main positive result.

**Theorem 1.** *When voters have beliefs that are any of the confident or unconfident versions of Mallows, Thurstone-Mosteller, or Placket-Luce under any parameter settings, plurality is OBIC.*

## 4 Other Voting Rules Are Not OBIC

From the positive result about plurality, one might wonder whether satisfying OBIC with respect to these beliefs is a relatively weak condition. If several rules satisfy it, this property is not useful for a mechanism designer who is comparing between rules to implement. In this section, we show this is not the case. Specifically, we focus on both the confident and unconfident variants of Mallows beliefs. Our main theoretical negative result is that plurality is uniquely OBIC among positional scoring rules in certain regimes of Mallows beliefs.

**Theorem 2.** *Let $f$ be a non-plurality positional scoring rule on three candidates. If a voter has unconfident or confident Mallows beliefs with $\varphi \leq .988$, for a sufficiently large $n$, $f$ is not OBIC.*

Below we provide a detailed proof sketch. However, we shunt some unwieldy technical derivations into two lemmas relegated to the appendix.

*Proof sketch of Theorem 2.* Fix a non-plurality scoring vector $(s_1, s_2, s_3)$. Without loss of generality, we can translate and scale the vector such that $s_3 = 0$ and $s_1$ and $s_2$ are relatively prime integers with $s_2 > 0$. Fix a voter $i$ with unconfident Mallows beliefs $\mathbb{P}$ with parameter $\varphi \leq .988$; we will describe later how to extend it to a confident Mallow's belief. Suppose they observe ranking $\hat{\sigma}_i = a \succ b \succ c$. We will show that for sufficiently large $n$, it will be beneficial to switch to $\sigma'_i = a \succ c \succ b$. More specifically, we will show that this manipulation increases the probability that candidate $a$ wins, which means the rule cannot be OBIC with respect to these beliefs. Formally, we will show that,

$$\Pr[f(\boldsymbol{\sigma}_{-i}, \sigma'_i) = a \mid \hat{\sigma}_i] > \Pr[f(\boldsymbol{\sigma}_{-i}, \hat{\sigma}_i) = a \mid \hat{\sigma}_i],$$

or equivalently,

$$\Pr[f(\boldsymbol{\sigma}_{-i}, \sigma_i') = a \mid \hat{\sigma}_i] - \Pr[f(\boldsymbol{\sigma}_{-i}, \hat{\sigma}_i) = a \mid \hat{\sigma}_i] > 0.$$

Notice that since we are looking at the difference in probabilities of two events, we can ignore their intersection when both reports lead to $a$ as the winner. That is, the left-hand side is equal to[3]

$$\Pr[f(\boldsymbol{\sigma}_{-i}, \sigma_i') = a \wedge f(\boldsymbol{\sigma}_{-i}, \hat{\sigma}_i) \neq a \mid \hat{\sigma}_i] - \Pr[f(\boldsymbol{\sigma}_{-i}, \hat{\sigma}_i) = a \wedge f(\boldsymbol{\sigma}_{-i}, \sigma_i') \neq a \mid \hat{\sigma}_i].$$

To shorten notation, we will label the events of interest as $\mathcal{E}^{cb} = \{f(\boldsymbol{\sigma}_{-i}, \sigma_i') = a \wedge f(\boldsymbol{\sigma}_{-i}, \hat{\sigma}_i) \neq a \mid \hat{\sigma}_i]\}$ (i.e., ranking $c$ above $b$ causes $a$ to win) and $\mathcal{E}^{bc} = \{f(\boldsymbol{\sigma}_{-i}, \hat{\sigma}_i) = a \wedge f(\boldsymbol{\sigma}_{-i}, \sigma_i') \neq a \mid \hat{\sigma}_i]\}$ (i.e., ranking $b$ above $c$ causes $a$ to win), so we wish to show that

$$\mathbb{P}[\mathcal{E}^{cb}|\hat{\sigma}_i] - \mathbb{P}[\mathcal{E}^{bc}|\hat{\sigma}_i] > 0.$$

Using the definition of the Mallows belief model, we can expand the left-hand side using ground truths to

$$\sum_{\tau \in \mathcal{L}} \left( \mathbb{P}[\mathcal{E}^{cb} \mid \tau] - \mathbb{P}[\mathcal{E}^{bc} \mid \tau] \right) \mathbb{P}[\tau \mid \hat{\sigma}_i]. \tag{2}$$

Consider the $\tau = a \succ c \succ b$ term. Notice that, by symmetry $\mathbb{P}[\mathcal{E}^{cb} \mid a \succ c \succ b] = \mathbb{P}[\mathcal{E}^{bc} \mid a \succ b \succ c]$ and $\mathbb{P}[\mathcal{E}^{bc} \mid a \succ c \succ b] = \mathbb{P}[\mathcal{E}^{cb} \mid a \succ b \succ c]$. Further $\mathbb{P}[a \succ c \succ b \mid \hat{\sigma}_i] = \varphi \cdot \mathbb{P}[a \succ b \succ c \mid \hat{\sigma}_i]$ as $d(a \succ c \succ b, \hat{\sigma}_i) = d(a \succ b \succ c, \hat{\sigma}_i) + 1$. Hence,

$$\left( \mathbb{P}[\mathcal{E}^{cb} \mid \tau = a \succ c \succ b] - \mathbb{P}[\mathcal{E}^{bc} \mid \tau = a \succ c \succ b] \right) \mathbb{P}[a \succ c \succ b \mid \hat{\sigma}_i]$$
$$= -\varphi \cdot \left( \left( \mathbb{P}[\mathcal{E}^{cb} \mid \tau = a \succ b \succ c] - \mathbb{P}[\mathcal{E}^{bc} \mid \tau = a \succ b \succ c] \right) \mathbb{P}[a \succ b \succ c \mid \hat{\sigma}_i] \right),$$

or in words, the $\tau = a \succ c \succ b$ term is exactly equal to $-\varphi$ times the $\tau = a \succ b \succ c$ term. In fact, this same property holds for the other two pairs of ground truth rankings where $a$ remains in the same position and $b$ is swapped with $c$, so $b \succ a \succ c$ with $c \succ a \succ b$ and $b \succ c \succ a$ with $c \succ b \succ a$. Hence, we can write the entire expression (2) as

$$(1 - \varphi) \cdot \Bigg( \left( \mathbb{P}[\mathcal{E}^{cb} \mid \tau = a \succ b \succ c] - \mathbb{P}[\mathcal{E}^{bc} \mid \tau = a \succ b \succ c] \right) \mathbb{P}[a \succ b \succ c \mid \hat{\sigma}_i]$$
$$+ \left( \mathbb{P}[\mathcal{E}^{cb} \mid \tau = b \succ a \succ c] - \mathbb{P}[\mathcal{E}^{bc} \mid \tau = b \succ a \succ c] \right) \mathbb{P}[b \succ a \succ c \mid \hat{\sigma}_i]$$
$$+ \left( \mathbb{P}[\mathcal{E}^{cb} \mid \tau = b \succ c \succ a] - \mathbb{P}[\mathcal{E}^{bc} \mid \tau = b \succ c \succ a] \right) \mathbb{P}[b \succ c \succ a \mid \hat{\sigma}_i] \Bigg).$$

Notice that since we wish to show this is strictly larger than $0$ and $1 - \varphi > 0$, we show only that the sum of the probability terms is positive. Additionally, subbing in the values of $\mathbb{P}[\tau \mid \hat{\sigma}_i]$ with the corresponding Kendall tau distances, the above simplifies to

$$\left( \mathbb{P}[\mathcal{E}^{cb} \mid \tau = a \succ b \succ c] - \mathbb{P}[\mathcal{E}^{bc} \mid \tau = a \succ b \succ c] \right)$$
$$+ \varphi \left( \mathbb{P}[\mathcal{E}^{cb} \mid \tau = b \succ a \succ c] - \mathbb{P}[\mathcal{E}^{bc} \mid \tau = b \succ a \succ c] \right) \tag{3}$$
$$+ \varphi^2 \left( \mathbb{P}[\mathcal{E}^{cb} \mid \tau = b \succ c \succ a] - \mathbb{P}[\mathcal{E}^{bc} \mid \tau = b \succ c \succ a] \right).$$

We will now show that for some $c_1 > c_2$ to be chosen later, the first positive term $\mathbb{P}[\mathcal{E}^{cb} \mid \tau = a \succ b \succ c] \in \Omega(c_1^n)$ and each negative term $\mathbb{P}[\mathcal{E}^{bc} \mid \tau] \in O(c_2^n)$, which implies that for sufficiently large $n$, the entire sum is positive, as needed. In addition, for the result to hold with confident Mallows rather than unconfident, it is only required that the first difference be positive, i.e.,

$$\mathbb{P}[\mathcal{E}^{cb} \mid \tau = a \succ b \succ c] - \mathbb{P}[\mathcal{E}^{bc} \mid \tau = a \succ b \succ c] > 0.$$

This is also directly implied by showing the above bounds.

We relegate these arguments to the following two lemmas, established in Appendices B and C, respectively.

**Lemma 3.** $\mathbb{P}[\mathcal{E}^{bc} \mid \tau] \in O(c_2^n)$ *for* $c_2 = e^{\frac{1-\varphi^2}{2(1+\varphi+\varphi^2)}} \sqrt{\frac{\varphi(1+2\varphi)}{1+\varphi+\varphi^2}}$.

---

[3]Note that since $f$ is randomized, to make this precise, we would need to specify the joint distribution of its outputs on different inputs. However, the remainder of the proof will not rely on how this is done, so the joint distribution can be arbitrary.

**Lemma 4.** $\mathbb{P}[\mathcal{E}^{cb} \mid \tau = a \succ b \succ c] \in \Omega(c_1^n)$ *for* $c_1 > e^{\frac{1-\varphi^2}{2(1+\varphi+\varphi^2)}}\sqrt{\frac{\varphi(1+2\varphi)}{1+\varphi+\varphi^2}}$.

Together, the two lemmas imply the desired result. $\qquad\square$

We now complement this result with some additional robustness checks. First, even though the result is asymptotic, in special cases of interest, this is, in fact, not necessary.

**Theorem 3.** *With three candidates, when a voter has unconfident or confident Mallows beliefs with* $\varphi < 1$*, Borda Count is not OBIC for* any $n \geq 2$.

Notice that $n = 1$ is a degenerate case with no other voters, so $n \geq 2$ is the strongest we can hope for. The proof of Theorem 3 can be found in Appendix D. The beginning is nearly identical to the proof sketch of Theorem 2, but they diverge after this point. While Lemmas 3 and 4 are asymptotic in nature, the corresponding portion for Theorem 3 requires careful counting of the number of profiles satisfying different conditions to ensure that for any fixed $n$, the inequalities hold.

In light of Theorem 3, it may seem plausible that Theorem 2 could be strengthened to hold for all $n$ rather than just asymptotically. However, we can give examples where this is not the case, suggesting that a "sufficiently large $n$" requirement may be necessary. In particular, there are scoring rules that, while not being plurality, are "close" to plurality in the sense that $s_2$ is so tiny it only matters when there is a tie among the plurality winners. For example, say we have the scoring rule $(4, 1, 0)$ with $n = 3$ voters. If any candidate receives two first-place votes, they immediately have $8$ out of the $15$ available points, so they necessarily win. Only when each candidate is ranked first by one voter is there any difference. We show in Appendix E that such close-to-plurality rules, at least for $n = 3$, are OBIC for confident Mallows beliefs with any $\varphi < 1$.

Finally, we consider other prominent non-scoring rules, namely Copeland and maximin. Note that for explicitly-defined beliefs and a number of voters $n$, we can determine whether or not a rule is OBIC by computing the probabilities of winners under all possible manipulations. We do so for the aforementioned rules under both confident and unconfident Mallows beliefs with $\varphi = 0.25, 0.5, 0.75$, $n = 2, \ldots, 50$, and $m = 3$. The results can be found in Table 2. Although slightly mixed in the sense that in a few specific cases, OBIC holds, the key takeaway is that none of the rules considered are as consistently OBIC as plurality.

|  | Copeland | maximin with $\varphi = 0.25, 0.5$ | maximin with $\varphi = 0.75$ |
|---|---|---|---|
| confident Mallows | even $n$ | $n \neq 3$ | $n \neq 3$ |
| unconfident Mallows | all | all | $n \neq 6$ |

Table 2: Scenarios where the Copeland and maximin rules *fail* to be OBIC with respect to Confident and Unconfident Mallows beliefs with $\varphi = 0.25, 0.50, 0.75$, $n = 2, \ldots 50$, and $m = 3$.

## 5 Discussion

In summary, we have considered the problem of strategic voting when voters have certain correlated beliefs over others. We have singled out plurality as an auspicious choice, being incentive compatible for a large class of beliefs, and have complemented this with negative results showing other prominent voting rules do not satisfy this property. However, our work is certainly not the final word on this topic. The current negative results are only for three candidates, and although we believe they should extend to a larger number, the technical work in showing this seems to get quite messy. Further, although we have checked many prominent voting rules, we have not ruled out the existence of other "reasonable" rules that perform as well as plurality while simultaneously satisfying other desiderata.

Finally, taking a more practical viewpoint, although OBIC is a theoretically compelling condition, it is susceptible to common criticisms of models of voter behavior. As with many models of this form, the utility difference of misreporting, or of choosing any vote for that matter, can be very low, and it is debatable whether this is the driving force in how voters make decisions. It raises questions similar in spirit to the so-called *Paradox of Voting* [6]: why would any rational agent choose to vote if the cost almost certainly outweighs the expected benefits? Despite these challenges, we do believe that the exploration and refinement of models such as OBIC can lead to an improved understanding of voter behavior and, ultimately, to the development of more effective voting systems.

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
