# Appendix

## A  Proof of Lemma 1

Fix a voter $i$, ranking $\hat{\sigma}_i$, and let $a = \hat{\sigma}_i(1)$ be their top choice. Fix an alternative $b \neq a$ and let $j^b = \hat{\sigma}_i^{-1}(b)$ be $b$'s position in $\hat{\sigma}_i$. Let $\mathbf{r}$ be a vector of plurality scores with $r_a > r_b$ and let $\mathbf{r}'$ be the same vector but with the $a$ and $b$ components swapped.

Let $\mathbb{P}$ be an unconfident Mallows, Placket-Luce, or Thurstone-Mosteller belief. We show later how this proof directly implies it for the confident version as well. We abuse notation slightly, and write $\mathbb{P}[\mathbf{r} \mid \hat{\sigma}_i]$ to denote the probability of the event that $\boldsymbol{\sigma}_{-i}$ has plurality vector $\mathbf{r}$, and $\mathbb{P}[\mathbf{r} \mid \tau]$ for the same under ground truth $\tau$. We wish to show that $\sum_\tau \mathbb{P}[\mathbf{r} \mid \tau] \cdot \mathbb{P}[\tau \mid \hat{\sigma}_i] \geq \sum_\tau \mathbb{P}[\mathbf{r}' \mid \tau] \cdot \mathbb{P}[\tau \mid \hat{\sigma}_i]$, or equivalently,

$$\sum_\tau (\mathbb{P}[\mathbf{r} \mid \tau] - \mathbb{P}[\mathbf{r}' \mid \tau]) \cdot \mathbb{P}[\tau \mid \hat{\sigma}_i] \geq 0.$$

Let $\tau$ be an arbitrary ground truth ranking with $a \succ_\tau b$ and let $\tau'$ be the same ranking, but with $a$ and $b$ switched. Notice that by symmetry, $\mathbb{P}[\mathbf{r} \mid \tau] = \mathbb{P}[\mathbf{r}' \mid \tau']$ and $\mathbb{P}[\mathbf{r}' \mid \tau] = \mathbb{P}[\mathbf{r} \mid \tau']$. Hence, in the above sum we can combine these two terms to be $(\mathbb{P}[\mathbf{r} \mid \tau] - \mathbb{P}[\mathbf{r}' \mid \tau])(\mathbb{P}[\tau \mid \hat{\sigma}_i] - \mathbb{P}[\tau' \mid \hat{\sigma}_i])$. We prove for all such $\tau$ with $a \succ_\tau b$, both of these terms are positive. Note that this immediately implies that this also holds for the confident version, as for that, we simply need to show $\mathbb{P}[\mathbf{r} \mid \tau] - \mathbb{P}[\mathbf{r}' \mid \tau]$ for $\tau = \hat{\sigma}_i$, and we have $a \succ_{\hat{\sigma}_i} b$.

We begin by showing $\mathbb{P}[\mathbf{r} \mid \tau] - \mathbb{P}[\mathbf{r}' \mid \tau] \geq 0$ for all $\tau$ with $a \succ_\tau b$. Fix such a $\tau$. Notice that, conditioned on $\tau$, all other rankings are drawn independently from the same distribution, namely, the corresponding model with ground truth $\tau$. Let $p_c$ be the probability that a ranking drawn from the corresponding model has top choice $c$. We can directly compute $\mathbb{P}[\mathbf{r} \mid \tau] = \binom{n-1}{\mathbf{r}} \prod_{c \in \mathcal{A}} p_c^{r_c}$ and $\mathbb{P}[\mathbf{r}' \mid \tau] = \binom{n-1}{\mathbf{r}'} \prod_{c \in \mathcal{A}} p_c^{r_c'}$, where $\binom{n-1}{\mathbf{r}}$ and $\binom{n-1}{\mathbf{r}'}$ are the multinomial coefficients, i.e., $\frac{(n-1)!}{\prod_{c \in \mathcal{A}} r_c!}$. To show the $\mathbb{P}[\mathbf{r} \mid \tau] \geq \mathbb{P}[\mathbf{r}' \mid \tau]$, observe that the two multinomial coefficients are equal as $\mathbf{r}$ and $\mathbf{r}'$ are the same up to swapping components. Further, since $r_c = r_c'$ for all $c \neq a, b$, the terms other than $a$ and $b$ are equal. Hence, all we need to show is that $p_a^{r_a} p_b^{r_b} \geq p_a^{r_a'} p_b^{r_b'}$. This will be directly implied by $p_a \geq p_b$.

For Mallow's, it is known that if $c = \tau(j)$, then the probability $c$ is the highest rank is proportional to $\varphi^j$. Hence, since $\tau^{-1}(a) < \tau^{-1}(b)$, $p_a > p_b$. For Placket-Luce, observe that each $p_c$ is proportional to $w_c$. Hence, $p_a > p_c$.

For Thurstone-Mosteller, things are more technical. Let $\mu_a > \mu_b$ be the corresponding means. We condition on arbitrary samples $x_c$ for $c \neq a, b$, and show that even conditioned on this, the probability $X_a$ is largest is greater than the probability that $X_b$ is largest. Since the conditioning was arbitrary, the law of total probability tells us that this is true in general.

Let $x_c^{max} = \max_{c \neq a,b} x_c$. Then, integrating over the standard normal PDF, the probability that $X_a$ is the largest is exactly

$$\int_{-\infty}^{\infty} \int_{-\infty}^{\infty} \frac{1}{2\pi} e^{-\frac{1}{2}(x-\mu_a)^2} e^{-\frac{1}{2}(y-\mu_b)^2} \mathbb{I}[x > \max(y, x_c^{max})] \, dx \, dy.$$

We can break up this integral depending on whether $X_b \geq x_c^{max}$ or not, to get that this is equal to

$$\frac{1}{2\pi} \left( \int_{-\infty}^{x_c^{max}} \int_{x_c^{max}}^{\infty} e^{-\frac{1}{2}(x-\mu_a)^2} e^{-\frac{1}{2}(y-\mu_b)^2} \, dx \, dy \right.$$
$$\left. + \int_{x_c^{max}}^{\infty} \int_{x_c^{max}}^{\infty} e^{-\frac{1}{2}(x-\mu_a)^2} e^{-\frac{1}{2}(y-\mu_b)^2} \mathbb{I}[x > y] \, dx \, dy \right).$$

The same can be done symmetrically for $X_b$. To show the probability is larger for $X_a$, we show that each of the terms is bigger, i.e.,

$$\int_{-\infty}^{x_c^{max}} \int_{x_c^{max}}^{\infty} e^{-\frac{1}{2}(x-\mu_a)^2} e^{-\frac{1}{2}(y-\mu_b)^2} \, dx \, dy \geq \int_{-\infty}^{x_c^{max}} \int_{x_c^{max}}^{\infty} e^{-\frac{1}{2}(x-\mu_b)^2} e^{-\frac{1}{2}(y-\mu_a)^2} \, dx \, dy$$

and

$$\int_{x_c^{max}}^{\infty} \int_{x_c^{max}}^{\infty} e^{-\frac{1}{2}(x-\mu_a)^2} e^{-\frac{1}{2}(y-\mu_b)^2} \mathbb{I}[x > y] \, dx \, dy$$

$$\geq \int_{x_c^{max}}^{\infty} \int_{x_c^{max}}^{\infty} e^{-\frac{1}{2}(x-\mu_b)^2} e^{-\frac{1}{2}(y-\mu_a)^2} \mathbb{I}[x > y] \, dx \, dy.$$

Both of these inequalities are implied by the fact that for all fixed $x > y$,

$$e^{-\frac{1}{2}(x-\mu_a)^2} e^{-\frac{1}{2}(y-\mu_b)^2} > e^{-\frac{1}{2}(x-\mu_b)^2} e^{-\frac{1}{2}(y-\mu_b)^2}.$$

Note that this is equivalent to showing

$$-\frac{1}{2}(x - \mu_a)^2 + -\frac{1}{2}(y - \mu_b)^2 \geq -\frac{1}{2}(x - \mu_b)^2 + -\frac{1}{2}(y - \mu_b)^2.$$

Indeed, we have that

$$-\frac{1}{2}((x - \mu_a)^2 + (y - \mu_b)^2) + \frac{1}{2}((x - \mu_b)^2 + (y - \mu_b)^2) = x\mu_a + y\mu_b - x\mu_b - y\mu_a$$
$$= (x - y)(\mu_a - \mu_b).$$

Since $x > y$ and $\mu_a > \mu_b$, this is positive, as needed.

Next, we wish to show $\mathbb{P}[\tau \mid \hat{\sigma}_i] \geq \mathbb{P}[\tau' \mid \hat{\sigma}_i]$. Recall that by Baye's rule, these are each proportional to $\mathbb{P}[\hat{\sigma}_i \mid \tau]$ and $\mathbb{P}[\hat{\sigma}_i \mid \tau']$, where these are the probabilities of drawing $\hat{\sigma}_i$ from the corresponding model with ground truth $\tau$ and $\tau'$. In the Mallows model, note $d(\hat{\sigma}_i, \tau) < d(\hat{\sigma}_i, \tau')$ because $a \succ_{\hat{\sigma}_i} b$, so swapping them can only increase the distance. Hence, $\mathbb{P}[\hat{\sigma}_i \mid \tau] \geq \mathbb{P}[\hat{\sigma}_i \mid \tau']$. For Placket-Luce, observe that the probability of generating a ranking $\sigma$ is

$$\prod_{j=1}^{m} \frac{w_{\sigma(j)}}{\sum_{j' \geq j} w_{\sigma(j')}}.$$

Notice that even reordering the weights $w$, the product of the numerators is always $\prod_{c \in \mathcal{A}} w_c$. However, the denominators can change. Let $w_a$ and $w_b$ be the weights of $a$ and $b$ under $\tau$, so $w_a > w_b$. The only difference between the denominators are those in terms with $j = 2, \ldots j^b$. Under $\tau$ these denominators include $w_b$ while under $\tau'$, this is replaced with $w_a$. Hence, under $\tau'$, all the denominators are at least as large, and hence the overall probability is less.

Finally, we handle Thurstone-Mosteller. Notice that under both ground truths $\tau$ and $\tau'$, $X_c$ for $c \neq a, b$ follow the same distributions. Hence, as before, we condition on values $x_c$ for $c \neq a, b$. If we show conditioned on any values, it is more likely to generate $\hat{\sigma}_i$ under $\tau$ then $\tau'$, then we are done. We first restrict to $x_c$ such that their order matches $\hat{\sigma}_i$, as otherwise, the probability of generating $\hat{\sigma}_i$ is 0. Again, let $x_c^{max} = \max_c x_c$. We now split into two cases based on if $j^b = 2$ or if $j^b > 2$. If $j^b = 2$. Then, the probability of generating $\hat{\sigma}_i$ under $\tau$ is

$$\int_{x_c^{max}}^{\infty} \int_{x_c^{max}}^{\infty} e^{-\frac{1}{2}(x-\mu_a)^2} e^{-\frac{1}{2}(y-\mu_b)^2} \mathbb{I}[x > y] \, dx \, dy.$$

Under $\tau'$, it is the same with $\mu_a$ and $\mu_b$ swapped. Similarly, when $j^b > 2$, then let $c^u = \hat{\sigma}_i(j^b - 1)$ and let $c^\ell = \hat{\sigma}_i(j^b + 1)$ be the candidates appearing directly before and after $b$ in $\hat{\sigma}_i$. The probability here of generating $\hat{\sigma}_i$ under $\tau$ is

$$\int_{x_{c^\ell}}^{x_{c^u}} \int_{x_c^{max}}^{\infty} e^{-\frac{1}{2}(x-\mu_a)^2} e^{-\frac{1}{2}(y-\mu_b)^2} \mathbb{I}[x > y] \, dx \, dy.$$

Under $\tau'$, it is again the same with $\mu_a$ and $\mu_b$ swapped. The proof that the $\tau$ versions are larger than the $\tau'$ follow the identical argument to the earlier ones showing $p_a > p_b$. $\qquad \square$

## B Proof of Lemma 3

Recall that $\mathcal{E}^{bc}$ is the event that $a$ wins in $(\boldsymbol{\sigma}_{-i}, \hat{\sigma}_i)$ but not $(\boldsymbol{\sigma}_{-i}, \sigma_i')$ where $\hat{\sigma}_i = abc$ and $\sigma_i' = acb$. Notice that in terms of scores, the only change when swapping from $\sigma_i'$ to $\hat{\sigma}_i$ is that $b$ has increased

by $r_1$ while $c$ has decreased by $r_1$. We claim that a necessary condition on $\boldsymbol{\sigma}_{-i}$ such that the probability of $a$ winning increases under this switch is that both $\mathrm{SC}_c(\boldsymbol{\sigma}_{-i}, \sigma_i') \geq \mathrm{SC}_a(\boldsymbol{\sigma}_{-i}, \sigma_i')$ and $\mathrm{SC}_a(\boldsymbol{\sigma}_{-i}, \hat{\sigma}_i) \geq \mathrm{SC}_b(\boldsymbol{\sigma}_{-i}, \hat{\sigma}_i)$. Indeed, if $\mathrm{SC}_c(\boldsymbol{\sigma}_{-i}, \sigma_i') < \mathrm{SC}_c(\boldsymbol{\sigma}_{-i}, \sigma_i')$, then even before the switch, $c$ was not a winning candidate, so decreasing their score and increasing $b$'s cannot improve $a$'s chances. Further, if $\mathrm{SC}_a(\boldsymbol{\sigma}_{-i}, \hat{\sigma}_i) < \mathrm{SC}_b(\boldsymbol{\sigma}_{-i}, \hat{\sigma}_i)$, then $a$ is winning with probability 0 on $(\boldsymbol{\sigma}_{-i}, \hat{\sigma}_i)$, so this cannot be an increase. Writing this only as a function of $\boldsymbol{\sigma}_{-i}$, we have that a necessary condition is that $\mathrm{SC}_c(\boldsymbol{\sigma}_{-i}) + r_2 \geq \mathrm{SC}_a(\boldsymbol{\sigma}_{-i}) + r_1$, $\mathrm{SC}_a(\boldsymbol{\sigma}_{-i}) + r_1 \geq \mathrm{SC}_b(\boldsymbol{\sigma}_{-i}) + r_2$, and (transitively from the previous two) $\mathrm{SC}_c(\boldsymbol{\sigma}_{-i}) \geq \mathrm{SC}_b(\boldsymbol{\sigma}_{-i})$.

We will show that for any $\tau = xyz$, $\mathbb{P}[\mathrm{SC}_z(\boldsymbol{\sigma}_{-i}) \geq \mathrm{SC}_x(\boldsymbol{\sigma}_{-i}) \mid \tau] \in O(c_2^n)$ (for $c_2$ to be chosen later). The above necessary conditions imply that this upper bounds each $\mathbb{P}[\mathcal{E}^{bc} \mid \tau]$ term.

To upperbound $\mathbb{P}[\mathrm{SC}_z(\boldsymbol{\sigma}_{-i}) \geq \mathrm{SC}_x(\boldsymbol{\sigma}_{-i}) \mid \tau]$, we will use a Chernoff bound. We begin by rewriting it as

$$\mathbb{P}[\mathrm{SC}_z(\boldsymbol{\sigma}_{-i}) - \mathrm{SC}_x(\boldsymbol{\sigma}_{-i}) \geq 0 \mid \tau] = \mathbb{P}\left[\sum_{j \neq i} \mathrm{SC}_z(\sigma_j) - \mathrm{SC}_x(\sigma_j) \geq 0 \,\middle|\, \tau\right].$$

Notice that conditioned on a ground truth $\tau$, each $\sigma_j$ (for $j \neq i$) is sampled independently from a Mallow's distribution around $\tau$. Hence, if we write $X_j = \mathrm{SC}_z(\sigma_j) - sc_x(\sigma_j)$, this is now the sum of independent random variables. To apply Chernoff, we will need that these are bounded between 0 and 1. As they are currently bounded in $[-r_1, r_1]$, we define $Y_j = \frac{X_j}{2r_1} + 1/2$, which is now bounded in $[0, 1]$. Hence, we wish to upperbound

$$\mathbb{P}\left[\frac{1}{n-1}\sum_{j \neq i} Y_j \geq 1/2 \,\middle|\, \tau\right]$$

To compute $\mathbb{E}[Y_j]$, we first compute $\mathbb{E}[X_j]$:

$$\mathbb{E}[X_j] = \sum_{\sigma \in \{xyz, xzy, yxz, zxy, yzx, zyx\}} (\mathrm{SC}_z(\sigma) - \mathrm{SC}_x(\sigma))\varphi^{d(\sigma,\tau)}$$

$$= \frac{(0 - r_1) \cdot 1 + (r_2 - r_1) \cdot \varphi + (0 - r_2) \cdot \varphi + (r_1 - r_2) \cdot \varphi^2 + (r_2 - 0) \cdot \varphi^2 + (r_1 - 0) \cdot \varphi^3}{1 + 2\varphi + 2\varphi^2 + \varphi^3}$$

$$= \frac{(-1 - \varphi + \varphi^2 + \varphi^3)r_1 + (\varphi - \varphi - \varphi^2 + \varphi^2)r_2}{1 + 2\varphi + 2\varphi^2 + \varphi^3}$$

$$= r_1 \cdot \frac{(1 + \varphi)(\varphi^2 - 1)}{(1 + \varphi)(1 + \varphi + \varphi^2)} = r_1 \cdot \frac{\varphi^2 - 1}{1 + \varphi + \varphi^2}.$$

From this we have that

$$\mathbb{E}[Y_j] = \frac{1}{2r_1}\mathbb{E}[X_j] + 1/2 = \frac{\varphi^2 - 1 + (1 + \varphi + \varphi^2)}{2(1 + \varphi + \varphi^2)} = \frac{\varphi(1 + 2\varphi)}{2(1 + \varphi + \varphi^2)}.$$

We will use the form of the Chernoff bound that states that if each $W_1, \ldots W_k$ is i.i.d. drawn from a distribution supported on $[0, 1]$ with $\mathbb{E}[W_j] = \mu$, then

$$\Pr\left[\frac{1}{k}\sum_j W_j \geq (1 + \delta)\mu\right] \leq \left(\frac{e^\delta}{(1 + \delta)^{1+\delta}}\right)^{k\mu} = \left(e^{(1+\delta)\mu - \mu}\left(\frac{\mu}{(1 + \delta)\mu}\right)^{(1+\delta)\mu}\right)^k. \tag{4}$$

Notice that in our case, $k = n - 1$, $\mu = \frac{\varphi(1+\varphi)}{2(1+\varphi+\varphi^2)}$, and $(1 + \delta)\mu = 1/2$. Hence, plugging in our values, we get that this is at most

$$\left(e^{\frac{1-\varphi^2}{2(1+\varphi+\varphi^2)}}\sqrt{\frac{\varphi(1 + 2\varphi)}{1 + \varphi + \varphi^2}}\right)^{n-1}.$$

Therefore, this quantity is $O(c_2^n)$ for $c_2 = e^{\frac{1-\varphi^2}{2(1+\varphi+\varphi^2)}}\sqrt{\frac{\varphi(1+2\varphi)}{1+\varphi+\varphi^2}}$. $\qquad\square$

# C Proof of Lemma 4

To lower bound $\mathbb{P}[\mathcal{E}^{cb} \mid \tau = abc]$, our strategy will be the following. First, we call a vector $\mathbf{h} = (h_\sigma)_{\sigma \in L}$ of integers indexed by $\mathcal{L}$ a *histogram*, and we will say that a profile $\boldsymbol{\sigma}$ has histogram $\mathbf{h}$ if $|\{i \mid \sigma_i = \sigma\}| = h_\sigma$. For all sufficiently large $n$, we will find histograms $(h_\sigma)_{\sigma \in L}$ with $\sum_{\sigma \in L} h_\sigma = n - 1$ such that on profiles $(\boldsymbol{\sigma}_{-i}, \sigma_i)$ with histogram $\mathbf{h}$, $a$ is tied with $b$ for the largest score, while on $(\boldsymbol{\sigma}_{-i}, \sigma'_i)$, $a$ is the unique winner. This implies that the probability $a$ wins for such profiles increases by at least $1/2$. We will then show that the probability that $\boldsymbol{\sigma}_{-i}$ has the corresponding histogram $h_\sigma$ is lower bounded by $\Omega(c_1^n)$.

To do this, we first must understand how likely it is to sample a profile with specific histogram $\mathbf{h}$. Let $p_\sigma = \varphi^{d(\sigma, a \succ b \succ c)} / Z$ be the probability of sampling $\sigma$ from the Mallow's distribution. Notice that sampling $\boldsymbol{\sigma}_{-i}$ and considering the counts $|\{i \in \boldsymbol{\sigma}_{-i} \mid \sigma_i = \sigma\}|$ is equivalent to drawing from a multinomial distribution over the alphabet $\mathcal{L}$ with probabilities $(p_\sigma)_{\sigma \in \mathcal{L}}$ of size $n - 1$. If we write $q_\sigma = h_\sigma / (n - 1)$ as the proportion of voters with $\sigma$, it is known that the probability of observing $(h_\sigma)_{\sigma \in \mathcal{L}}$ (with each $h_\sigma > 0$) is at least $\left(\prod_\sigma \left(\frac{p_\sigma}{q_\sigma}\right)^{q_\sigma} - o(1)\right)^{n-1}$. Note that $\prod_\sigma \left(\frac{p_\sigma}{q_\sigma}\right)^{q_\sigma} = 1/e^{D_{KL}(\mathbf{p}\|\mathbf{q})}$ where $D_{KL}$ is the KL-divergence and $\mathbf{p}$ and $\mathbf{q}$ are treated as probability distributions over $\mathcal{L}$. This is essentially (without uniform convergence) an immediate consequence of the tightness of Sanov's theorem [21], although it can easily be derived by known bounds on multinomial coefficients [4].

With this property in hand, we now wish to find profiles satisfying the tie conditions such that $\left(\frac{p_\sigma}{q_\sigma}\right)^{q_\sigma}$ is bounded away from $e^{\frac{1-\varphi^2}{2(1+\varphi+\varphi^2)}} \sqrt{\frac{\varphi(1+2\varphi)}{1+\varphi+\varphi^2}}$. To that end, we now show the following:

**Lemma 5.** *For all $\varphi \leq .988$ and positional scoring rules $(r_1, r_2, 0)$, there exists real numbers $(q_\sigma)_{\sigma \in L}$ such that:*

1. *They are valid proportions: $\sum_\sigma q_\sigma = 1$ and each $q_\sigma > 0$.*

2. *Candidates $a$ and $b$ are tied in score: $\sum_\sigma \mathrm{sc}_a(\sigma) q_\sigma = \sum_\sigma sc_b(\sigma) q_\sigma$.*

3. *Candidate $c$ is not beating $a$ and $b$: $\sum_\sigma \mathrm{sc}_a(\sigma) q_\sigma \geq \sum_\sigma sc_c(\sigma) q_\sigma$.*

4. *The objective of these $q$'s are large $\prod_\sigma \left(\frac{p_\sigma}{q_\sigma}\right)^{q_\sigma} > e^{\frac{1-\varphi^2}{2(1+\varphi+\varphi^2)}} \sqrt{\frac{\varphi(1+2\varphi)}{1+\varphi+\varphi^2}}$.*

*Proof.* We first handle the case where $\varphi \leq 0.1$. Under this assumption of $\varphi$, we can explicitly choose $q_\sigma$ as follows.

$$q_{abc} = q_{bac} = \frac{\sqrt{p_{abc}p_{bac}}}{2(\sqrt{p_{abc}p_{bac}} + \sqrt{p_{acb}p_{bca}} + \sqrt{p_{cab}p_{cba}})}$$

$$q_{acb} = q_{bca} = \frac{\sqrt{p_{acb}p_{bca}}}{2(\sqrt{p_{abc}p_{bac}} + \sqrt{p_{acb}p_{bca}} + \sqrt{p_{cab}p_{cba}})}$$

$$q_{cab} = q_{cba} = \frac{\sqrt{p_{cab}p_{cba}}}{2(\sqrt{p_{abc}p_{bac}} + \sqrt{p_{acb}p_{bca}} + \sqrt{p_{cab}p_{cba}})}.$$

Since all $p_\sigma$ are positive, each $q_\sigma$ is positive. Further, they are explicitly chosen to add up to one. In addition, due to the symmetry between $a$ and $b$ (they appear in each position at the same frequency), their corresponding scores are equal. Finally, since $p_{abc} > p_{bac} \geq p_{acb} > p_{bca} \geq p_{cab} > p_{cba}$, it follows that $q_{abc} = q_{bac} > q_{acb} = q_{bca} > q_{cab} = q_{cba}$, so the score of $c$ is strictly less than the score of $a$. It remains to be shown that $\prod_\sigma \left(\frac{p_\sigma}{q_\sigma}\right)^{q_\sigma} > e^{\frac{1-\varphi^2}{2(1+\varphi+\varphi^2)}} \sqrt{\frac{\varphi(1+2\varphi)}{1+\varphi+\varphi^2}}$. Let $d = 2(\sqrt{p_{abc}p_{bac}} + \sqrt{p_{acb}p_{bca}} + \sqrt{p_{cab}p_{cba}})$ be the denominator in each of the $q$ values. Let us consider the contribution to the product of the $abc$ and $bac$ terms. We have,

$$\left(\frac{p_{abc}}{q_{abc}}\right)^{q_{abc}} \cdot \left(\frac{p_{bac}}{q_{bac}}\right)^{q_{bac}} = \left(\frac{dp_{abc}}{\sqrt{p_{abc}p_{bca}}}\right)^{q_{abc}} \cdot \left(\frac{dp_{bac}}{\sqrt{p_{abc}p_{bca}}}\right)^{q_{bac}}$$

$$= d^{q_{abc}+q_{bac}} \cdot \left( \frac{p_{abc}}{\sqrt{p_{abc}p_{bca}}} \cdot \frac{p_{bac}}{\sqrt{p_{abc}p_{bca}}} \right)^{q_{abc}}$$

$$= d^{q_{abc}+q_{bca}} \cdot 1$$

The same argument holds for the other two pairs, which implies that

$$\prod_\sigma \left( \frac{p_\sigma}{q_\sigma} \right)^{q_\sigma} = d^{\sum_\sigma q_\sigma} = d.$$

Expanding the value of $d$,

$$2\left( \sqrt{\frac{\varphi^0 \cdot \varphi^1}{Z^2}} + \sqrt{\frac{\varphi^1 \cdot \varphi^2}{Z^2}} + \sqrt{\frac{\varphi^2 \cdot \varphi^3}{Z^2}} \right) = \frac{2\sqrt{\varphi}(1+\varphi+\varphi^2)}{Z}$$

$$= \frac{2\sqrt{\varphi}(1+\varphi+\varphi^2)}{1+2\varphi+2\varphi^2+\varphi^3}$$

$$= \frac{2\sqrt{\varphi}}{1+\varphi}.$$

Finally, using the assumption that $\varphi \le .1$, we have

$$\frac{2\sqrt{\varphi}}{1+\varphi} \ge \frac{2\sqrt{\varphi}}{1.1}$$

$$= \sqrt{e \cdot 1.2} \cdot \sqrt{\varphi}$$

$$\ge e^{1/2} \cdot \sqrt{\varphi(1+2\varphi)}$$

$$> (e^{1/2})^{\frac{1-\varphi^2}{1+\varphi+\varphi^2}} \cdot \frac{1}{\sqrt{1+\varphi+\varphi^2}} \cdot \sqrt{\varphi(1+2\varphi)}$$

$$= e^{\frac{1-\varphi^2}{2(1+\varphi+\varphi^2)}} \sqrt{\frac{\varphi(1+2\varphi)}{1+\varphi+\varphi^2}},$$

where the second inequality uses the fact that $2/1.1 \approx 1.82 > \sqrt{1.2e} \approx 1.81$.

Next, we consider $\varphi > 0.1$. We formalize finding valid $q$s in the following form. Notice first that we can rescale the scoring vector to be of the form $(1, \alpha, 0)$ where $\alpha = r_2/r_1 \in [0, 1]$. We will use $\text{sc}_x^\alpha(\sigma)$ to denote the score of candidate $x$ on ranking $\sigma$ with the positional scoring rule $(1, \alpha, 0)$. Let $Q_\alpha$ be the set of vectors $\mathbf{q}$ (indexed by $\mathcal{L}$), which satisfy the constraints for a specific $\alpha$. Expanding the objective in terms of $\varphi$, let $d_\sigma = d(\sigma, a \succ b \succ c)$, $f(\varphi, \mathbf{q}) = \frac{1}{1+2\varphi+2\varphi^2+\varphi^3} \prod_\sigma \left( \frac{\varphi^{d_\sigma}}{q_\sigma} \right)^{q_\sigma}$, and $\ell(\varphi) = e^{\frac{1-\varphi^2}{2(1+\varphi+\varphi^2)}} \sqrt{\frac{\varphi(1+2\varphi)}{1+\varphi+\varphi^2}}$. Let $g(\varphi, \mathbf{q}) = f(\varphi, \mathbf{q}) - \ell(\varphi)$. Our goal is to show that for all $\varphi \in (.1, .99]$ and for all $\alpha \in [0, 1]$, there is a $\mathbf{q} \in Q_\alpha$ such that $g(\varphi, \mathbf{q}) > 0$. When $\mathbf{q}$ satisfies this, we will say that $\mathbf{q}$ is a *solution* for $\varphi$ and $\alpha$.

To that end, we will first show using the smoothness of $g$ and the $Q_\alpha$ sets that as long as a solution $\mathbf{q}$ for a specific $\varphi$ and $\alpha$ satisfies reasonable conditions, then that will imply the existence of solutions for nearby $\varphi$ and $\alpha$. We will then present several solutions found using a computational search that cover the $\alpha$ and $\varphi$ region, implying the existence of solutions for all necessary values.

Fix $\alpha$, $\varphi$, and suppose we have a corresponding solution $\mathbf{q}$. Fix some $\varepsilon > 0$, we now find sufficient conditions such that for all $\varphi' \in [\varphi - \varepsilon, \varphi + \varepsilon]$ and $\alpha' \in [\alpha - \varepsilon, \alpha + \varepsilon]$, there exists a solution $\mathbf{q}'$ for $\varphi'$ and $\varepsilon'$. We begin by extending it to the same $\alpha$, but for $\varphi' \in [\varphi - \varepsilon, \varphi + \varepsilon]$. We first show that $\ell$ is an increasing function on $[0, 1]$ which implies (as long as $\varphi + \varepsilon \le 1$), on $[\varphi - \varepsilon, \varphi + \varepsilon]$, it is upper bounded by $\ell(\varphi + \varepsilon)$.

Indeed, notice that $\mu$ (from (4)) is equal to $\frac{\varphi+2\varphi^2}{2(1+\varphi+\varphi^2)} = 1/2 - \frac{1-\varphi^2}{2(1+\varphi+\varphi^2)}$ and its derivative with respect to $\varphi$ is $\frac{\varphi^2+4\varphi+1}{(\varphi^2+\varphi+1)^2} > 0$. Therefore, it is an increasing function of $\varphi$ bounded in $[0, 1/2]$. As a function of $\mu$, $\ell(\varphi)$ is equal to $(e/2)^{1/2}e^{-\mu}\sqrt{\mu}$. The derivative of this with respect to $\mu$ is

$(e/2)^{1/2}e^{-\mu}(1-2\mu)/(2\sqrt{\mu})$, positive for $\mu \in [0, 1/2)$. Therefore, as the composition of two increasing functions, $\ell(\varphi)$ is increasing on $[0, 1]$.

Next, we wish to lower bound $f(\varphi', \mathbf{q})$. To do this, suppose the derivative $\frac{\partial f}{\partial \varphi}(\varphi', \mathbf{q})$ for $\varphi' \in [\varphi - \varepsilon, \varphi + \varepsilon]$ lower bounded by $B \leq 0$. Notice that $-B$ upper bounds the rate at which $f$ can decrease, so we get that $f(\varphi', \mathbf{q}) \geq f(\varphi - \varepsilon, \mathbf{q}) + 2\varepsilon B$. To compute such a $B$, we first compute $\frac{\partial f}{\partial \varphi}(\varphi', \mathbf{q})$. We will use the fact that $\frac{\partial f}{\partial \varphi}(\varphi', \mathbf{q}) = \frac{\partial \log(f)}{\partial \varphi}(\varphi', \mathbf{q}) \cdot f(\varphi', \mathbf{q})$. Since $\log(f(\varphi', \mathbf{q})) = \sum_\sigma q_\sigma(d_\sigma \log(\varphi') - q\sigma) - \log(1 + 2\varphi' + 2\varphi'^2 + \varphi'^3)$, we have that

$$\frac{\partial f}{\partial \varphi}(\varphi', \mathbf{q}) = \left( \frac{\sum_\sigma q_\sigma d_\sigma}{\varphi'} - \frac{2 + 4\varphi' + 3\varphi'^2}{1 + 2\varphi' + 2\varphi'^2 + \varphi'^3} \right) \cdot f(\varphi', \mathbf{q}).$$

Notice that $\frac{\sum_\sigma q_\sigma d_\sigma}{\varphi'}$ is decreasing in $\varphi'$. Further, we can also show that $\frac{2+4\varphi'+3\varphi'^2}{1+2\varphi'+2\varphi'^2+\varphi'^3}$ is decreasing, as its derivative is

$$-\frac{\varphi'(3\varphi'^3 + 8\varphi'^2 + 8\varphi' + 2)}{(\varphi'^3 + 2\varphi'^2 + 2\varphi' + 1)^2},$$

negative for all positive values of $\varphi'$. Finally, notice that $f$ is defined as $1/e^{D_{KL}(\mathbf{q}\|\mathbf{p})}$ and $D_{KL}$ is nonnegative, $f$ is upperbounded by 1. Hence, for all $\varphi' \in [\varphi - \varepsilon, \varphi + \varepsilon]$,

$$\frac{\partial f}{\partial \varphi}(\varphi', \mathbf{q}) \geq \min\left( \frac{\sum_\sigma q_\sigma d_\sigma}{\varphi + \varepsilon} - \frac{2 + 4(\varphi - \varepsilon) + 3(\varphi - \varepsilon)^2}{1 + 2(\varphi - \varepsilon) + 2(\varphi - \varepsilon)^2 + (\varphi - \varepsilon)^3}, 0 \right).$$

Let $B(\varphi, \mathbf{q}, \varepsilon) = \min\left( \frac{\sum_\sigma q_\sigma d_\sigma}{\varphi + \varepsilon} - \frac{2 + 4(\varphi - \varepsilon) + 3(\varphi - \varepsilon)^2}{1 + 2(\varphi - \varepsilon) + 2(\varphi - \varepsilon)^2 + (\varphi - \varepsilon)^3}, 0 \right)$. We then have that for all $\varphi' \in [\varphi - \varepsilon, \varphi + \varepsilon]$, $g(\varphi', q) \geq f(\varphi - \varepsilon, \mathbf{q}) + 2\varepsilon B(\varphi, \mathbf{q}, \varepsilon) - \ell(\varphi + \varepsilon, \mathbf{q})$.

Next, we consider modifying $\alpha$ to $\alpha' \in [\alpha - \varepsilon, \alpha + \varepsilon]$. Let $\beta = \alpha' - \alpha$. Notice that the current $\mathbf{q}$ may not be an element of $Q_{\alpha'}$. Although $\sum_\sigma q_\sigma = 1$, and each $q_\sigma > 0$, it may not be the case $\sum_\sigma \text{sc}_b^{\alpha'}(\sigma)q_\sigma = \sum_\sigma \text{sc}_a^{\alpha'}(\sigma)q_\sigma$. Instead, we have that $\sum_\sigma \text{sc}_b^{\alpha'}(\sigma)q_\sigma + \beta(q_{abc} + q_{cba}) = \sum_\sigma \text{sc}_a^{\alpha'}(\sigma)q_\sigma + \beta(q_{bac} + q_{cab})$. Let $r = q_{abc} + q_{cba} - q_{bac} - q_{cab}$; this is the current amount $b$ is beating $a$ by (it may be negative). Notice that we can find a $\mathbf{q}' \in Q_{\alpha'}$ by simply shifting $r/2 \cdot \beta$ mass from $q_{acb}$ to $q_{bca}$. This will result in a valid $\mathbf{q}'$ as long as $q_{acb} > r/2 \cdot \beta$ when $r/2 \cdot \beta$ is positive or $q_{bca} > -r/2 \cdot \beta$ when it is negative. A sufficient condition for this is that both $q_{acb} > |r\varepsilon/2|$ and $q_{bca} > |r\varepsilon/2|$. Under this assumption, we now consider the effect on the solution value $g(\varphi, \mathbf{q}')$. To do this, we can consider the directional derivative of $g$ with respect to increasing $q_{acb}$ and decreasing $q_{bca}$. We have that for each $\sigma$,

$$\frac{\partial g}{\partial q_\sigma} = f(\varphi, \mathbf{q}) \cdot \left( \log\left( \frac{\varphi^{d_\sigma}}{q_\sigma} \right) - 1 \right).$$

Therefore, the derivative with respect the the vector of increasing $q_{acb}$ and dcreasing $q_{bca}$ is

$$\frac{\partial g}{\partial q_{acb}} - \frac{\partial g}{\partial q_{bca}} = f(\varphi, \mathbf{q}) \cdot \left( \log(\varphi^{d_{acb} - d_{bca}}) + \log\left( \frac{q_{bca}}{q_{acb}} \right) \right) = f(\varphi, \mathbf{q}) \cdot \left( \log\left( \frac{q_{bca}}{q_{acb}} \right) - \log(\varphi) \right).$$

We will now upperbound the magnitude of this. Recall that $f$ is upper bounded by 1. Further, for any $\varphi' \in [\varphi + \varepsilon, \varphi - \varepsilon]$ and $\mathbf{q}'$ constructed by shifting at most $r\varepsilon/2$ mass between $q_{acb}$ and $q_{bca}$,

$$\log\left( \frac{q_{bca} - |r\varepsilon/2|}{q_{acb} + |r\varepsilon/2|} \right) - \log(\varphi + \varepsilon) \leq \log\left( \frac{q_{bca}}{q_{acb}} \right) - \log(\varphi) \leq \log\left( \frac{q_{bca} + |r\varepsilon/2|}{q_{acb} - |r\varepsilon/2|} \right) - \log(\varphi - \varepsilon).$$

Hence, the magnitude of the derivative is always at most:

$$\max\left( \left| \log\left( \frac{q_{bca} - |r\varepsilon/2|}{q_{acb} + |r\varepsilon/2|} \right) - \log(\varphi + \varepsilon) \right|, \left| \log\left( \frac{q_{bca} + |r\varepsilon/2|}{q_{acb} - |r\varepsilon/2|} \right) - \log(\varphi - \varepsilon) \right| \right).$$

Let $m(\varphi, \mathbf{q}, \varepsilon)$ be this value. Then, from shifting the at most $r\varepsilon/2$ mass between $q_{acb}$ and $q_{bca}$, this decreases $g(\varphi, \mathbf{q})$ by at most $r\varepsilon/2 \cdot m(\varphi, \mathbf{q}, \varepsilon)$. Hence, putting this all together, we have that for any vector $\mathbf{q}$, as long as both $q_{acb}, q_{bca} > r\varepsilon/2$, and as long as

$$f(\varphi - \varepsilon, \mathbf{q}) + 2\varepsilon B(\varphi, \mathbf{q}, \varepsilon) - \ell(\varphi + \varepsilon, \mathbf{q}) - \frac{r\varepsilon}{2}m(\varphi, \mathbf{q}, \varepsilon) > 0,$$

then this implies that for all $\varphi' \in [\varphi - \varepsilon, \varphi + \varepsilon]$ and $\alpha' \in [\alpha - \varepsilon, \alpha + \varepsilon]$, there exists a solution $\mathbf{q}'$.

Finally, for all $0.1 \leq \varphi \leq .988$ and $0 \leq \alpha \leq 1$ that are multiples of $1/1000$, we compute corresponding $\mathbf{q}$ that satisfy the above conditions with $\varepsilon = 1/2000$. Together, these cover the space of $\varphi$ and $\alpha$, which implies that the lemma holds. This can be done (approximately enough) using a convex program to find $\mathbf{q}$ that maximizes $f$ given $\varphi$ and $\alpha$. The computed values can be found in the supplementary material. $\qquad\square$

Notice that the solutions $(q_\sigma)_{\sigma \in \mathcal{L}}$ from Lemma 5 need not be rational which would be necessary for a valid profile with corresponding $(h_\sigma)_{\sigma \in \mathcal{L}}$ to be sampled. However, we claim that given a non-rational solution, we can always find a rational one, so it is without loss of generality to assume they are. Notice that since the strict inequalities are all continuous functions of $q$, so there must be an $\varepsilon > 0$ such that all $q$ vectors in an $\varepsilon$-ball around these $q$s (in $\mathbb{R}^6$) satisfy the strict inequalities. In addition, the linear equalities form an affine subspace. Since all coefficients are rational, all-rational vectors are dense within this subspace. Hence, there are rational $(q'_\sigma)_{\sigma \in \mathcal{L}}$ within $\varepsilon$ of $(q_\sigma)_{\sigma \in L}$ that satisfies the equalities and is, therefore, a rational solution to the four properties.

Using rational $\mathbf{q}$, we can find a corresponding integral $\mathbf{h}$ such that on profiles with ranking counts equal to $\mathbf{h}$, $a$ and $b$ are tied for winning. Let $s = \sum_\sigma h_\sigma$ be the number of rankings in $\mathbf{h}$. For a ranking $\sigma$, let $\mathbf{e}_\sigma$ be the unit vector with 1 in the $\sigma$ coordinate and 0 elsewhere. Notice that if $n - 1 = ks + 1$ for some integer $k$ and $\boldsymbol{\sigma}_{-i}$ has ranking counts equal to $k\mathbf{h} + \mathbf{e}_{bac}$, then it is indeed the case that on $(\boldsymbol{\sigma}_{-i}, a \succ b \succ c)$, $a$ is tied with $b$, while on $(\boldsymbol{\sigma}_{-i}, a \succ c \succ b)$, $a$ is the unique winner.

To handle cases where $n - 2$ is not a multiple of $s$, suppose we write $n - 1 = k \cdot h + 1 + r$ where $2 \leq r \leq s + 1$. If $r$ is odd, we can first add a cycle $\mathbf{e}_{abc} + \mathbf{e}_{bca} + \mathbf{e}_{cab}$ which does not affect relative scores. After doing this, we can add $r/2$ (or $(r - 3)/2$ if $r$ was odd) copies of $\mathbf{e}_{abc} + \mathbf{e}cab$ which again keeps $a$ and $b$ at the same relative scores and only pushes $c$ down. By doing this, we can get a histogram of arbitrary size where $(\boldsymbol{\sigma}_{-i}, a \succ b \succ c)$ has $a$ tied with $b$ and $(\boldsymbol{\sigma}_{-i}, a \succ c \succ b)$ has $a$ as a unique winner. Finally, notice that $n$ grows large, the proportion of this histogram approaches $\mathbf{q}$. Hence, for sufficiently large $n$, the probability of sampling this histogram will be $\Omega(c_1^n)$ for any $c_1 < \prod_\sigma \left( \frac{p_\sigma}{q_\sigma} \right)^{q_\sigma}$. Since $\prod_\sigma \left( \frac{p_\sigma}{q_\sigma} \right)^{q_\sigma} > c_2$, we can choose $c_1 > c_2$. This completes the proof. $\qquad\square$

## D  Proof of Theorem 3

Consider the Borda scoring rule $(2, 1, 0)$, and a voter $i$ with unconfident Mallows belief $\mathbb{P}$ with $\varphi < 1$. As usual, we will describe how to extend it to confident Mallows later. The proof begins identically to Theorem 2, up to the point of needing to show (3) is nonnegative. We restate (3) here for convenience.

$$
\begin{aligned}
&\left( \mathbb{P}[\mathcal{E}^{cb} \mid \tau = a \succ b \succ c] - \mathbb{P}[\mathcal{E}^{bc} \mid \tau = a \succ b \succ c] \right) \\
&+ \varphi \left( \mathbb{P}[\mathcal{E}^{cb} \mid \tau = b \succ a \succ c] - \mathbb{P}[\mathcal{E}^{bc} \mid \tau = b \succ a \succ c] \right) \\
&+ \varphi^2 \left( \mathbb{P}[\mathcal{E}^{cb} \mid \tau = b \succ c \succ a] - \mathbb{P}[\mathcal{E}^{bc} \mid \tau = b \succ c \succ a] \right).
\end{aligned}
$$

Here, we show each of the probability differences are nonnegative, and the first is strictly positive. This also implies that the result holds for confident Mallows where only the first strict inequality is necessary.

To do this, we provide an equivalent way of computing $\mathbb{P}[\mathcal{E}^{cb} \mid \tau] - \mathbb{P}[\mathcal{E}^{bc} \mid \tau]$. Let us consider the profiles $\boldsymbol{\sigma}_{-i}$ where swapping from $a \succ b \succ c$ to $a \succ c \succ b$ leads to an increase in the probability $a$ wins. Notice that the swap decreases the score of $b$ by 1 and increases the score of $c$ by 1. For this to help $a$ win, $b$ must have been one of the winners before. Therefore, one of the following must hold.

1. On $(\boldsymbol{\sigma}_{-i}, \sigma_i)$, $a$ was tied with $b$ with $c$ being at least two behind them. Then, after the swap, $a$ wins outright, an increase in winning probability of $1/2$.

2. On $(\boldsymbol{\sigma}_{-i}, \sigma_i)$, $b$ was winning outright, $a$ was one point behind, and $c$ was more than one point behind $a$, then, after the swap, $a$ and $b$ are tied winners, an increase in winning probability of $1/2$.

3. On $(\boldsymbol{\sigma}_{-i}, \sigma_i)$, $b$ was winning outright, $a$ was one point behind, and $c$ was one point behind $a$, then, after the swap, all three are tied, an increase of winning probability of $1/3$.

We define sets $A_1, A_2, A_3$ of profiles $\boldsymbol{\sigma}_{-i}$ that correspond to these three events. More formally,

$$A_1 = \{\boldsymbol{\sigma}_{-i} \mid \mathrm{SC}_b(\boldsymbol{\sigma}_{-i}) = \mathrm{SC}_a(\boldsymbol{\sigma}_{-i}) + 1 \geq \mathrm{SC}_c(\boldsymbol{\sigma}_{-i}) + 1\},$$
$$A_2 = \{\boldsymbol{\sigma}_{-i} \mid \mathrm{SC}_b(\boldsymbol{\sigma}_{-i}) = \mathrm{SC}_a(\boldsymbol{\sigma}_{-i}) + 2 \geq \mathrm{SC}_c(\boldsymbol{\sigma}_{-i}) + 2\},$$
$$A_3 = \{\boldsymbol{\sigma}_{-i} \mid \mathrm{SC}_b(\boldsymbol{\sigma}_{-i}) = \mathrm{SC}_a(\boldsymbol{\sigma}_{-i}) + 2 = \mathrm{SC}_c(\boldsymbol{\sigma}_{-i}) + 1\}.$$

We can analogously define $B_1$, $B_2$, and $B_3$ with $b$ and $c$ swapped, which correspond to profiles where swapping causes the probability of $a$ winning to decrease. From this, we get that

$$P[\mathcal{E}^{cb} \mid \tau] - \mathbb{P}[\mathcal{E}^{bc} \mid \tau] = \frac{1}{2}\mathbb{P}[A_1 \mid \tau] + \frac{1}{2}\mathbb{P}[A_2 \mid \tau] + \frac{1}{3}\mathbb{P}[A_3 \mid \tau]$$
$$- \left(\frac{1}{2}\mathbb{P}[B_1 \mid \tau] + \frac{1}{2}\mathbb{P}[B_2 \mid \tau] + \frac{1}{3}\mathbb{P}[B_3 \mid \tau]\right)$$

Further, notice that there is a natural bijection $\pi$ between the sets of profiles $A_k$ and $B_k$ for $k \leq 3$, namely, swapping every occurrence of $b$ with $c$ and vice-versa.

To prove a weak inequality, we will show that for each $k$ and each $\tau$, $\mathbb{P}[A_k \mid \tau] \geq \mathbb{P}[B_k \mid \tau]$. Notice that this is simply a statement about draws of profiles from a Mallows model; voter $i$ and their report do not have an impact. To make calculations less messy we will simply refer to these profiles without $i$ as $\boldsymbol{\sigma}$ instead of $\boldsymbol{\sigma}_{-i}$ and refer to the set of voters $V$ as the ones without $i$, and the size of these profiles as $n$ (even though this is technically $n-1$). This means that our assumption now is that $n \geq 1$.

Next, as a simplifying step, fix an arbitrary partition of the voters $K_1, K_2, K_3$, i.e., $V = K_1 \sqcup K_2 \sqcup K_3$. Let

$$A_k^{K_1,K_2,K_3} = \{\boldsymbol{\sigma}_{-i} \in A_k \mid \sigma_j(1) = a, \forall j \in K_1 \wedge \sigma_j(2) = a, \forall j \in K_2 \wedge \sigma_j(3) = a \forall j \in K_3\}.$$

In words $A_k^{K_1,K_2,K_3}$ is the subset of $A_k$ such that the voters in $K_1$ rank $a$ first, voters in $K_2$ rank $a$ second, and voters in $K_3$ rank $a$ third. We define this analogously for the $B_k$ sets. We will show for all partitions $K_1, K_2, K_3$, $\mathbb{P}[A_k^{K_1,K_2,K_3} \mid \tau] \geq \mathbb{P}[B_k^{K_1,K_2,K_3} \mid \tau]$ which implies it holds for the original sets.

Fix an arbitrary $K_1, K_2, K_3$ and $k \leq 3$. Writing this out more explicitly and using the $\pi$ bijection, we see that it suffices to show for each $\tau$,

$$\sum_{\boldsymbol{\sigma} \in A_k^{K_1,K_2,K_3}} (\varphi^{d(\boldsymbol{\sigma},\tau)} - \varphi^{d(\pi(\boldsymbol{\sigma}),\tau)}) \geq 0. \tag{5}$$

We assume now that $A_k^{K_1,K_2,K_3} \neq \emptyset$ as otherwise this inequality trivially holds.

Notice that for all $\boldsymbol{\sigma} \in A_k^{K_1,K_2,K_3}$,

$$\mathrm{SC}_a(\boldsymbol{\sigma}) = 2|K_1| + |K_2| \tag{6}$$

In other words, the score of $a$ on all profiles in $A_k^{K_1,K_2,K_3}$ is constant. From this, we can derive the scores of the other candidates.

$$\mathrm{SC}_b(\boldsymbol{\sigma}) = \mathrm{SC}_a(\boldsymbol{\sigma}) + \kappa = 2|K_1| + |K_2| + \kappa. \tag{7}$$

where $\kappa = 1, 2$ depending on whether $k = 1$ or $k \in \{2, 3\}$. Finally, for all $\boldsymbol{\sigma}$, $\mathrm{SC}_a(\boldsymbol{\sigma}) + \mathrm{SC}_b(\boldsymbol{\sigma}) + \mathrm{SC}_c(\boldsymbol{\sigma}) = 3n$. Therefore,

$$\mathrm{SC}_a(\boldsymbol{\sigma}) = 3n - \mathrm{SC}(a) - sc(b) = 3n - 4|K_1| - 2|K_2| - \kappa. \tag{8}$$

Further, these equations are an equivalent condition for defining $A_k^{K_1,K_2,K_3}$, a profile $\boldsymbol{\sigma} \in A_k^{K_1,K_2,K_3}$ if and only if the voters in each of $K_1$, $K_2$, and $K_3$ rank $a$ accordingly and Equations (6) to (8) are all satisfied.

Additionally, we have that for any $\boldsymbol{\sigma} \in A_1 \cup A_2 \cup A_3$, $\mathrm{SC}_c(\boldsymbol{\sigma}) \leq \mathrm{SC}_b(\boldsymbol{\sigma}) - 1$. Therefore, by the assumption that $A_k^{K_1,K_2,K_3}$ was nonempty, we can derive some constraints on $|K_1|, |K_2|$, and $|K_3|$. Namely, for any $\boldsymbol{\sigma} \in A_k^{K_1,K_2,K_3}$,

$$3(|K_1| + |K_2| + |K_3|) = 3n$$

$$= \mathrm{sc}_a(\boldsymbol{\sigma}) + \mathrm{sc}_b(\boldsymbol{\sigma}) + \mathrm{sc}_c(\boldsymbol{\sigma})$$
$$= \mathrm{sc}_a(\boldsymbol{\sigma}) + 2\mathrm{sc}_b(\boldsymbol{\sigma}) - 1$$
$$= 6|K_1| + 3|K_2| + 2\kappa - 1.$$

Therefore,

$$|K_1| \geq |K_3| - \frac{2\kappa - 1}{3}. \tag{9}$$

Recall that for a pair of candidates $x$ and $y$, $N_{xy}(\boldsymbol{\sigma}) = |\{i | x \succ_i y\}|$. Note that $N_{bc}(\boldsymbol{\sigma}) = n - N_{bc}(\pi(\boldsymbol{\sigma}))$ since all occurrences of $b$ and $c$ are swapped. It can be shown that for Borda scores,

$$\mathrm{sc}_x(\boldsymbol{\sigma}) = \sum_{y \neq x} N_{xy}(\boldsymbol{\sigma}) \tag{10}$$

In addition, by the definition of the Kendall tau distance,

$$d(\boldsymbol{\sigma}, xyz) = N_{yx}(\boldsymbol{\sigma}) + N_{zx}(\boldsymbol{\sigma}) + N_{zy}(\boldsymbol{\sigma}), \tag{11}$$

as this counts the total number of swapped pairs.

We now handle the cases of each $\tau \in \{abc, bac, cba\}$ separately.

**Case 1: $\tau = abc$.** By Equations (10) and (11), we have that

$$d(\boldsymbol{\sigma}, abc) = N_{ba}(\boldsymbol{\sigma}) + N_{ca}(\boldsymbol{\sigma}) + N_{cb}(\boldsymbol{\sigma})$$
$$= n - N_{ab}(\boldsymbol{\sigma}) + n - N_{ac}(\boldsymbol{\sigma}) + n - N_{bc}(\boldsymbol{\sigma})$$
$$= 2n - \mathrm{sc}_a(\boldsymbol{\sigma}) + (n - N_{bc}(\boldsymbol{\sigma}))$$

and

$$d(\pi(\boldsymbol{\sigma}), abc) = N_{ba}(\pi(\boldsymbol{\sigma})) + N_{ca}(\pi(\boldsymbol{\sigma})) + N_{cb}(\pi(\boldsymbol{\sigma}))$$
$$= 2n - \mathrm{sc}_a(\pi(\boldsymbol{\sigma})) + (n - N_{bc}(\pi(\boldsymbol{\sigma})))$$
$$= 2n - \mathrm{sc}_a(\boldsymbol{\sigma}) + N_{bc}(\boldsymbol{\sigma}).$$

Substituting this into the left-hand side of (5), we have

$$\sum_{\boldsymbol{\sigma} \in A_k^{K_1, K_2, K_3}} \varphi^{d(\boldsymbol{\sigma}, abc)} - \varphi^{d(\pi(\boldsymbol{\sigma}), abc)}$$

$$= \sum_{\boldsymbol{\sigma} \in A_k^{K_1, K_2, K_3}} \varphi^{2n - \mathrm{sc}_a(\boldsymbol{\sigma}) + n - N_{bc}(\boldsymbol{\sigma})} - \varphi^{2n - \mathrm{sc}_a(\boldsymbol{\sigma}) + N_{bc}(\boldsymbol{\sigma})}$$

$$= \sum_{\boldsymbol{\sigma} \in A_k^{K_1, K_2, K_3}} \varphi^{2n - 2|K_1| - |K_2|} \left( \varphi^{n - N_{bc}(\boldsymbol{\sigma})} - \varphi^{N_{bc}(\boldsymbol{\sigma})} \right)$$

Note that the term in front is always nonnegative and constant for fixed $K_1, K_2, K_3$, so it is sufficient to show

$$\sum_{\boldsymbol{\sigma} \in A_k^{K_1, K_2, K_3}} \left( \varphi^{n - N_{bc}(\boldsymbol{\sigma})} - \varphi^{N_{bc}(\boldsymbol{\sigma})} \right) \geq 0. \tag{12}$$

Notice that these terms depend only on $N_{bc}(\boldsymbol{\sigma})$ which must take on a value in $\{0, \ldots, n\}$. Hence, we can instead consider counting the number of profiles $\boldsymbol{\sigma} \in A_k^{K_1, K_2, K_3}$ with a specific $N_{bc}(\boldsymbol{\sigma})$. More formally, let $Q_j = |\{\boldsymbol{\sigma} \in A_k^{K_1, K_2, K_3} | N_{bc}(\boldsymbol{\sigma}) = j\}|$ for $j \in \{0, \ldots, n\}$. We can now write

$$\sum_{\boldsymbol{\sigma} \in A_k^{K_1, K_2, K_3}} \left( \varphi^{n - N_{bc}(\boldsymbol{\sigma})} - \varphi^{N_{bc}(\boldsymbol{\sigma})} \right) = \sum_{j=0}^{n} Q_j \left( \varphi^{n-j} - \varphi^j \right).$$

Notice that for each $Q_j(\varphi^{n-j} - \varphi^j)$ term in the sum, there is a corresponding term $Q_{n-j}(\varphi^j - \varphi^{n-j})$. Pairing up these opposite terms, we can rewrite the sum as

$$\sum_{j=0}^{\lfloor (n-1)/2 \rfloor} (Q_{n-j} - Q_j) \left( \varphi^j - \varphi^{n-j} \right)$$

The $\lfloor (n-1)/2 \rfloor$ expression is simply the largest integer strictly less than $n/2$ (we exclude the $j = n/2$ term since this is $0$ if it exists). Note that $(\varphi^j - \varphi^{n-j}) > 0$ for $j < n/2$, so we have reduced the problem to counting the number of profiles $\boldsymbol{\sigma}$ with a specific value of $N_{bc}(\boldsymbol{\sigma})$. More formally, Inequality (12) to show for $j < n/2$,

$$Q_j \leq Q_{n-j}. \tag{13}$$

Fix a $j < n/2$. For a profile $\boldsymbol{\sigma} \in A_k^{K_1,K_2,K_3}$, define

$$t(\boldsymbol{\sigma}) = \{i \in K_2 | b \succ_i c\}$$
$$o(\boldsymbol{\sigma}) = \{i \in K_1 \cup K_3 | b \succ_i c\},$$

In words, $t(\boldsymbol{\sigma})$ is the number of voters in $K_2$ that prefer $b$ to $c$ and $o(\boldsymbol{\sigma})$ is the number of voters in $K_1 \cup K_3$ that prefer $b$ to $c$. This is useful for us because these values allow us to calculate $\mathrm{SC}_b(\boldsymbol{\sigma})$. Voters in $|K_3|$ give a minimum of one point to $b$. For all voters counted in $o(\boldsymbol{\sigma})$, an additional one point is given versus those not counted. For all voters counted in $t(\boldsymbol{\sigma})$ an additional two points are given versus those not counted. Hence,

$$\mathrm{SC}_b(\boldsymbol{\sigma}) = |K_3| + 2t(\boldsymbol{\sigma}) + o(\boldsymbol{\sigma}).$$

When $\boldsymbol{\sigma} \in A_k^{K_1,K_2,K_3}$, we know that $\mathrm{SC}_b(\boldsymbol{\sigma}) = \mathrm{SC}_a(\boldsymbol{\sigma}) + \kappa$, so we have that

$$2t(\boldsymbol{\sigma}) + o(\boldsymbol{\sigma}) = 2|K_2| + |K_1| + \kappa - |K_3| \tag{14}$$

Further,

$$t(\boldsymbol{\sigma}) + o(\boldsymbol{\sigma}) = N_{bc}(\boldsymbol{\sigma})$$

as it is simply a different way of counting the number of voters with $b \succ c$.

Observe that if $\boldsymbol{\sigma}$ is counted toward $Q_j$, both Equation (14) must hold and $t(\boldsymbol{\sigma}) + o(\boldsymbol{\sigma}) = j$. These are two independent linear equations on $t(\boldsymbol{\sigma})$ and $o(\boldsymbol{\sigma})$ and hence there is exactly one solution for $t(\boldsymbol{\sigma})$ and $o(\boldsymbol{\sigma})$ that satisfies them. Further, notice that this is a necessary and sufficient condition: $\boldsymbol{\sigma} \in A_k^{K_1,K_2,K_3}$ is counted toward $Q_j$ if and only if it satisfies both Equation (14) and $t(\boldsymbol{\sigma}) + o(\boldsymbol{\sigma}) = j$ (along with the $K_1, K_2, K_3$ constraint).

Let $t$ and $o$ be the solutions satisfying the above equations for $Q_j$ with $0 \leq t \leq |K_2|$ and $0 \leq o \leq |K_1| + |K_3|$. Note that if $t$ or $o$ are not integers or do not satisfy the inequalities then $Q_j = 0$, so $Q_j \leq Q_{n-j}$ as $Q_{n-j}$ is necessarily nonnegative. For $t$ and $o$ satisfying the inequalities, we have that

$$Q_j = \binom{|K_2|}{t}\binom{|K_1| + |K_3|}{o}$$

since we choose $t$ voters in $|K_2|$ and $o$ voters in $|K_1| \cup |K_3|$ to rank $b \succ c$. We first claim that $t' := t - n + 2j$ and $o' := o + 2n - 4j$ are solutions for $Q_{n-j}$ since

$$2t' + o' = 2(t - n + 2j) + (o + 2n - 4j) = 2t + o = 2|K_2| + |K_1| + \kappa - |K_3|$$

$$t' + o' = t - n + 2j + o + 2n - 4j = t + o + n - 2j = j + n - 2j = n - j$$

Since $t$ and $o$ were integers, so are $t'$ and $o'$. We want to show

$$Q_j = \binom{|K_2|}{t}\binom{|K_1| + |K_3|}{o} \leq \binom{|K_2|}{t'}\binom{|K_1| + |K_3|}{o'} = Q_{n-j}$$

We will show individually that $\binom{|K_2|}{t'} \geq \binom{|K_2|}{t}$ and $\binom{|K_1|+|K_3|}{o'} \geq \binom{|K_1|+|K_3|}{o}$. Notice that $t' \leq t$ and $o' \geq o$, so this is implied by showing that $t' \geq |K_2| - t$ and $o' \leq |K_1| + |K_3| - o$. Both rely on the inequality $2(2t + o) > n + |K_2|$, which follows from

$$\begin{aligned}
2(2t + o) &= 2(2|K_1| + |K_2| + \kappa - |K_3|) \\
&= 4|K_1| + 2|K_2| - 2|K_3| + 2\kappa \\
&\geq |K_1| + 2|K_2| + 3|K_3| - 2|K_3| + 2\kappa - (2\kappa - 1) && (|K_1| \geq |K_3| - \tfrac{2\kappa - 1}{3}) \\
&\geq n + |K_2| && (|K_1| + |K_2| + |K_3| = n)
\end{aligned}$$

Using the derived inequality, we have

$$t' = t - n + 2j$$

$$\begin{aligned}
&= t - n + 2(t + o) \\
&= 3t + 2o - n \\
&= 2(2t + o) - n - t \\
&\geq n + |K_2| - n - t \\
&= |K_2| - t
\end{aligned}$$

and

$$\begin{aligned}
o' &= o + 2n - 4N \\
&= o + 2n - 4(o + t) \\
&= 2n - 3o - 4t \\
&= 2n - 2(o + 2t) - o \\
&\leq 2n - (n - |K_2|) - o \\
&= n - |K_2| - o \\
&= |K_1| + |K_3| - o, \hspace{4cm} (|K_1| + |K_2| + |K_3| = n)
\end{aligned}$$

Therefore, Inequality (13) holds, as needed.

**Case 2:** $\tau = bca$. Again, by Equations (10) and (11), we have that

$$\begin{aligned}
d(\boldsymbol{\sigma}, bca) &= N_{ab}(\boldsymbol{\sigma}) + N_{ac}(\boldsymbol{\sigma}) + N_{cb}(\boldsymbol{\sigma}) \\
&= \mathrm{SC}_a(\boldsymbol{\sigma}) + n - N_{bc}(\boldsymbol{\sigma})
\end{aligned}$$

and

$$\begin{aligned}
d(\pi(\boldsymbol{\sigma}), bca) &= N_{ab}(\pi(\boldsymbol{\sigma})) + N_{ac}(\pi(\boldsymbol{\sigma})) + N_{cb}(\pi(\boldsymbol{\sigma})) \\
&= \mathrm{SC}_a(\pi(\boldsymbol{\sigma})) + (n - N_{bc}(\pi(\boldsymbol{\sigma}))) \\
&= \mathrm{SC}_a(\boldsymbol{\sigma}) + N_{bc}(\boldsymbol{\sigma}).
\end{aligned}$$

Substituting this into the left-hand side of (5), we have,

$$\sum_{\boldsymbol{\sigma} \in A_k^{K_1,K_2,K_3}} \varphi^{d(\boldsymbol{\sigma},bca)} - \varphi^{d(\pi(\boldsymbol{\sigma}),bca)} = \sum_{\boldsymbol{\sigma} \in A_k^{K_1,K_2,K_3}} \varphi^{\mathrm{SC}_a(\boldsymbol{\sigma})+(n-N_{bc}(\boldsymbol{\sigma}))} - \varphi^{\mathrm{SC}_a(\boldsymbol{\sigma})+N_{bc}(\boldsymbol{\sigma})}$$

$$= \sum_{\boldsymbol{\sigma} \in A_k^{K_1,K_2,K_3}} \varphi^{2|K_1|+|K_2|} \left( \varphi^{n-N_{bc}(\boldsymbol{\sigma})} - \varphi^{N_{bc}(\boldsymbol{\sigma})} \right).$$

Again we notice that the term in front is always nonnegative and constant for fixed $K_1, K_2, K_3$, so, it is sufficient to show

$$\sum_{\boldsymbol{\sigma} \in A_k^{K_1,K_2,K_3}} \left( \varphi^{n-N_{bc}(\boldsymbol{\sigma})} - \varphi^{N_{bc}(\boldsymbol{\sigma})} \right) \geq 0,$$

which we already proved in the last case.

**Case 3:** $\tau = bac$. Using Equations (10) and (11), we have

$$\begin{aligned}
d(\boldsymbol{\sigma}, bac) &= N_{ab}(\boldsymbol{\sigma}) + N_{ca}(\boldsymbol{\sigma}) + N_{cb}(\boldsymbol{\sigma}) \\
&= (n - N_{ba}(\boldsymbol{\sigma})) + (n - N_{ac}(\boldsymbol{\sigma})) + (n - N_{bc}(\boldsymbol{\sigma})) \\
&\quad + \underbrace{(n - N_{ab}(\boldsymbol{\sigma}) - N_{ba}(\boldsymbol{\sigma}))}_{0} + \underbrace{(N_{bc}(\boldsymbol{\sigma}) - N_{bc}(\boldsymbol{\sigma}))}_{0} \\
&= 4n - \mathrm{SC}_a(\boldsymbol{\sigma}) - 2\mathrm{SC}_b(\boldsymbol{\sigma}) + N_{bc}(\boldsymbol{\sigma}) \\
&= 4n - 6|K_1| - 3|K_2| - 2\kappa + N_{bc}(\boldsymbol{\sigma}) \\
&= (-2|K_1| + |K_2| + 4|K_3| - 2\kappa) + N_{bc}(\boldsymbol{\sigma})
\end{aligned}$$

Similarly,

$$\begin{aligned}
d(\pi(\boldsymbol{\sigma}), bac) &= N_{ab}(\pi(\boldsymbol{\sigma})) + N_{ca}(\pi(\boldsymbol{\sigma})) + N_{cb}(\pi(\boldsymbol{\sigma})) \\
&= (n - N_{ab}(\boldsymbol{\sigma})) + (n - N_{ca}(\boldsymbol{\sigma})) + (n - N_{cb}(\boldsymbol{\sigma}))
\end{aligned}$$

$$= 3n - d(\boldsymbol{\sigma}, bac)$$
$$= 3n - (-2|K_1| + |K_2| + 4|K_3| - 2\kappa) - N_{bc}(\boldsymbol{\sigma})$$

Let $C = (-2|K_1| + |K_2| + 4|K_3| - 2\kappa)$. Substituting this into the left-hand side of (5), we have .

$$\sum_{\boldsymbol{\sigma} \in A_k^{K_1, K_2, K_3}} \varphi^{d(\boldsymbol{\sigma}, bac)} - \varphi^{d(\pi(\boldsymbol{\sigma}), cab)} = \sum_{\boldsymbol{\sigma} \in A_k^{K_1, K_2, K_3}} \varphi^{C + N_{bc}(\boldsymbol{\sigma})} - \varphi^{3n - C - N_{bc}(\boldsymbol{\sigma})}$$

$$= \sum_{j=0}^{n} Q_j \left( \varphi^{C+j} - \varphi^{3n-C-j} \right).$$

Observe that $\left( \varphi^{C+j} - \varphi^{3n-C-j} \right)$ is negative only for $j > \frac{3n}{2} - C$. Additionally, for each of these terms, there is a corresponding positive term in the sum for $j' = 3n - 2C - j < \frac{3n}{2} - C$, where

$$\left( \varphi^{C+j'} - \varphi^{3n-C-j'} \right) = \left( \varphi^{C+3n-2C-j} - \varphi^{3n-C-3n+2C+j} \right) = - \left( \varphi^{C+j} - \varphi^{3n-C-j} \right).$$

Thus, it suffices to show for $j > \frac{3n}{2} - C$ that $Q_j \leq Q_{j'}$ where $j' = 3n - 2C - j$.

As before, let $t$ and $o$ be solutions for $Q_j$. Then we have the following solutions for $Q_{j'}$

$$t' = t + (j - j')$$
$$o' = o - 2(j - j')$$

since

$$t' + o' = t + (j - j') + o - 2(j - j') = j'$$
$$2t' + o' = 2t + 2(j + j')o - 2(j - j') = 2t + o.$$

Recall that
$$Q_j = \binom{|K_2|}{t} \binom{|K_1| + |K_3|}{o} \text{ and } Q_{j'} = \binom{|K_2|}{t'} \binom{|K_1| + |K_3|}{o'}.$$

We will show that $\binom{|K_2|}{t} \leq \binom{|K_2|}{t'}$ and $\binom{|K_1| + |K_3|}{o} \leq \binom{|K_1| + |K_3|}{o'}$. Note that $t' \geq t$ and $o' \leq o$, so it suffices to show that $t' \leq |K_2| - t$ and $o' \geq |K_1| + |K_3| - o$. Let us directly consider

$$t' = t + (j - j')$$
$$= t + (2j - 3n + 2C)$$
$$= t + (2(t + o) - 3n + 2C)$$
$$= 2(2t + o) - 3n + 2C - t$$
$$= 2(2|K_1| + |K_2| + \kappa - |K_3|) - 3n + 2(-2|K_1| + |K_2| + 4|K_3| - 2\kappa) - t$$
$$= -3n + 4|K_2| + 6|K_3| - 2\kappa - t$$
$$= -3|K_1| + 3|K_3| - 2\kappa + |K_2| - t$$
$$< |K_2| - t.$$

We also have that

$$o' = o - 2(j - j')$$
$$= o - 2(2j - 3n + 2C)$$
$$= o - 2(2(t + o) - 3n + 2C)$$
$$= -2(2t + o) + 6n - 4C - o$$
$$= -2(2|K_1| + |K_2| + \kappa - |K_3|) + 6n - 4(-2|K_1| + |K_2| + 4|K_3| - 2\kappa) - o$$
$$= 6n + 4|K_1| - 6|K_2| - 14|K_3| + 6\kappa - o$$
$$= 10|K_1| - 8|K_3| + 6\kappa - o$$
$$= 9|K_1| - 9|K_3| + 6\kappa + |K_1| + |K_3| - o$$
$$> |K_1| + |K_3| - o.$$

| Report | Probability $a$ wins |
|--------|----------------------|
| $abc$ | $1\varphi^0 + 4\varphi^1 + 7\varphi^2 + 8\varphi^3 + 8/3\varphi^4 + 0\varphi^5 + 0\varphi^6$ |
| $bac$ | $1\varphi^0 + 2\varphi^1 + 3\varphi^2 + 2\varphi^3 + 2/3\varphi^4 + 0\varphi^5 + 0\varphi^6$ |
| $acb$ | $1\varphi^0 + 4\varphi^1 + 7\varphi^2 + 8\varphi^3 + 8/3\varphi^4 + 0\varphi^5 + 0\varphi^6$ |
| $bca$ | $1\varphi^0 + 2\varphi^1 + 5/3\varphi^2 + 0\varphi^3 + 0\varphi^4 + 0\varphi^5 + 0\varphi^6$ |
| $cab$ | $1\varphi^0 + 4\varphi^1 + 11/3\varphi^2 + 0\varphi^3 + 0\varphi^4 + 0\varphi^5 + 0\varphi^6$ |
| $cba$ | $1\varphi^0 + 2\varphi^1 + 5/3\varphi^2 + 0\varphi^3 + 0\varphi^4 + 0\varphi^5 + 0\varphi^6$ |

| Report | Probability $c$ wins |
|--------|----------------------|
| $abc$ | $0\varphi^0 + 0\varphi^1 + 0\varphi^2 + 0\varphi^3 + 5/3\varphi^4 + 2\varphi^5 + 1\varphi^6$ |
| $bac$ | $0\varphi^0 + 0\varphi^1 + 0\varphi^2 + 0\varphi^3 + 5/3\varphi^4 + 2\varphi^5 + 1\varphi^6$ |
| $acb$ | $0\varphi^0 + 0\varphi^1 + 0\varphi^2 + 0\varphi^3 + 11/3\varphi^4 + 4\varphi^5 + 1\varphi^6$ |
| $bca$ | $0\varphi^0 + 0\varphi^1 + 2/3\varphi^2 + 2\varphi^3 + 3\varphi^4 + 2\varphi^5 + 1\varphi^6$ |
| $cab$ | $0\varphi^0 + 0\varphi^1 + 8/3\varphi^2 + 8\varphi^3 + 7\varphi^4 + 4\varphi^5 + 1\varphi^6$ |
| $cba$ | $0\varphi^0 + 0\varphi^1 + 8/3\varphi^2 + 8\varphi^3 + 7\varphi^4 + 4\varphi^5 + 1\varphi^6$ |

Table 3: Probability that $a$ and $c$ each win under different reports for voter $i$. This assumes their observed ranking was $abc$.

This completes the proof for the weak inequality.

To show that the first inequality is strict, observe that in case 1, all of the inequalities about $Q_j \leq Q_{n-j}$ can be shown to be strict. Hence, all we need to show is that there is some $k$ such that $A_k$ is nonempty. Fix some arbitrary $n$. If $n$ is even, we can take a profile $\boldsymbol{\sigma}$ where $n/2 + 1$ voters have the ranking $bac$ and $n/2 - 1$ have $abc$. Such a profile is always an element of $A_2$ since $\mathrm{SC}_b(\boldsymbol{\sigma}) = \mathrm{SC}_a(\boldsymbol{\sigma}) + 2$ and $\mathrm{SC}_a(\boldsymbol{\sigma}) \geq \mathrm{SC}_c(\boldsymbol{\sigma})$. Similarly, if $n$ is odd, we can take a profile $\boldsymbol{\sigma}$ where $\lceil n/2 \rceil$ voters have the ranking $bac$ and $\lfloor n/2 \rfloor$ have the ranking $acb$. Such a profile is always an element of $A_1$ since $\mathrm{SC}_b(\boldsymbol{\sigma}) = \mathrm{SC}_a(\boldsymbol{\sigma}) + 1$ and $\mathrm{SC}_a(\boldsymbol{\sigma}) \geq \mathrm{SC}_c(\boldsymbol{\sigma})$. $\qquad\square$

# E   OBIC Positional Scoring Rule Example

Let $f$ be a scoring rule $(r_1, r_2, 0)$ such that $r_2/r_1 < 1/3$. Note that on three voters, all these rules coincide. Indeed, if there is a strict plurality winner, that candidate is necessarily the winner. If not, this means each candidate appeared first exactly once. I some candidate appears in second twice, then that candidate is the winner. Finally, if no candidate appears in second twice, they all appear in second once, and therefore all appear in third once, so there is a three-way tie. Under such rules with a confident Mallows prior with a fixed $\varphi$, we can explicitly compute the probability that each candidate wins as a function of $\varphi$. This assumes, without loss of generality, that the voter's observed ranking is $abc$. The probability that $a$ and $c$ each win under possible reports are shown in Table 3. One can check that reporting anything other than $abc$ neither increases the probability that $a$ wins or decreases the probability that $c$ wins, which means the rule is OBIC.