# OpenReview forum: "Strategyproof Voting under Correlated Beliefs"
_NeurIPS.cc/2023/Conference — NeurIPS 2023 poster_

### Official Review · Reviewer_A4JJ · 2023-07-04

**Soundness:** 3 good
**Presentation:** 3 good
**Contribution:** 3 good
**Rating:** 5
**Confidence:** 4

**Summary:**

Post rebuttal: I improved my scores and encourage the authors to include the information from the rebuttal in the final text.

The authors study strategy-proofness in voting under the assumption that the voters do not have, as is usual, full knowledge about the votes/preferences of the other voters, but rather have some form of beliefs about them. Specifically, they consider that the election they participate in was generated using the Mallows model and either their vote is the central one (confident setting) or not. They also consider several other similar distributions. Their main result is that in the setting, Plurality is strategy-proof.

On the mathematical level the paper is OK, but I have serious doubts about the significance of the result. In short, the assumed belief models mean that from the perspective of the considered agent, she is winning. Indeed, if the central ballot agrees with the current vote, the the top choice candidate is expected to have the highest plurality score. Then, strategy-proofness for Plurality is essentially built-in into the definition.  The situation where strategy proofness and strategic behavior is interesting is where there is expected contention. The current model does not capture this.

Another issue is that, as far as I can tell, the message from the paper would be to use the Plurality rule. But this certainly is not a rule we would like to use.

All in all, the paper certainly makes a step in an interesting direction and, perhaps, this direction would eventually lead to interesting results. However, in the current shape, it is quite far from giving any definite answers (and, as such, can certainly find home in more focused and specialized conferences).

**Strengths:**

makes some progress in a widely studied topic

**Weaknesses:**

- the results are not relevant practically
- the strategyproofness for Plurality is, in essence, built into the considered model

**Questions:**

Would you consider your results as arguments for using the Plurality rule in practice?

**Limitations:**

Limitations were addressed

---

> ### Author Rebuttal · Authors · 2023-08-03
>
>
> **Reviewer comment:**
>
> >  In short, the assumed belief models mean that from the perspective of the considered agent, she is winning. Indeed, if the central ballot agrees with the current vote, the the top choice candidate is expected to have the highest plurality score. Then, strategy-proofness for Plurality is essentially built-in into the definition. The situation where strategy proofness and strategic behavior is interesting is where there is expected contention. The current model does not capture this.
>
> **Response:**
>
> This seems to be the reviewer's main concern. We are confident that it stems from a misunderstanding, and believe the rebuttal should fully address it.
>
> First, the reviewer appears to be referring to the "confident" versions of our model, whereas our results also apply to the "unconfident" versions.
>
> Second, and much more importantly, the statement "the top choice candidate is expected to have the highest plurality score" is also true when "plurality" is replaced with, say, "Borda" — but our results show that Borda is not OBIC in our setting! The question is not whether the manipulator's top choice has the highest chance of winning (not to mention the highest expected score), but whether reporting a different ranking leads to this candidate winning *even more often*. It is not the case that one implies the other. Indeed, almost all rules satisfy the first, but, of the ones we study, only plurality avoids the second. In summary, the reviewer's justification for the assertion that "strategy-proofness for Plurality is essentially built-in into the definition" is incorrect.
>
> **Reviewer comment:**
>
> > Another issue is that, as far as I can tell, the message from the paper would be to use the Plurality rule. But this certainly is not a rule we would like to use. [...] Would you consider your results as arguments for using the Plurality rule in practice?
>
> **Response:**
>
> We disagree with the statement that plurality "is not a rule we would like to use": with few exceptions, it's the voting rule that's always used in practice.
>
> Now, based on anecdotal evidence, many social choice theorists do seem to prefer other voting rules to plurality. But this preference is based on a variety of criteria for comparing voting rules: axiomatic desiderata, maximum likelihood estimation under various noise models, distortion, distance rationalizability, etc. Simplicity of preference elicitation is another important criterion, in which plurality excels.
>
> Strategyproofness typically isn't a primary criterion in the comparison of voting rules, due to Gibbard-Satterthwaite. Our approach and results can be seen as potentially elevating this criterion so that it can augment the set of criteria that are being examined. Whether this would tilt the scales in favor of plurality is subjective, but we believe it's a criterion that should certainly play a role in the discussion.
>
> To explicitly answer your question: yes, we view our results as arguments for using plurality in practice, which should be considered alongside other theoretical and empirical arguments.

---

> > ### Comment · Reviewer_A4JJ · 2023-08-18
> > **Response**
> >
> > I accept your arguments and I like that you were bold enough to clarify your view on Plurality. I think the paper would benefit from making them clear in the text as well (although, admittingly, researchers will challenge your view).
> >
> > I am still not overly excited with the work, but sufficiently to not choose rejecting evaluations.

---

> > > ### Author Response · Authors · 2023-08-18
> > >
> > > We greatly appreciate your attention to our rebuttal. We'll be sure to incorporate the points raised in the rebuttal into the revised version of the paper.

---

### Official Review · Reviewer_kNTn · 2023-07-06

**Soundness:** 3 good
**Presentation:** 2 fair
**Contribution:** 2 fair
**Rating:** 5
**Confidence:** 2

**Summary:**

This paper explores a probabilistic form of strategy-proofness for voting rules referred to as Ordinally Bayesian Incentive Compatible (OBIC) in a setting where voters believe that other voters have correlated votes. The paper considers both the situation when voters believe others have preferences similar to their own, and when they believe other voters have correlated preferences that are not necessarily similar to their own. The primary positive result of the paper explores three correlated preference/belief models and shows that they are "top-choice correlated"; ie. under them, when all else is equal a "voter's top choice is likely to perform better than other candidates." It is then shown that under any top-choice correlated beliefs the plurality rule is OBIC. Subsequent negative results identify a number of specific situations where other scoring rules are not OBIC.

**Strengths:**

The paper is generally well written. Theorems are clearly stated and there is a strong combination of high level explanation alongside technical definitions. Reasonable motivation is given for studying the questions focused on in the paper, and it is already a well-established area of research.


While there is a great deal of research on problems of this nature under independent beliefs, thus far the body of work examining results under correlated beliefs is comparatively small and much more recent. This paper adds novel results to the domain. The significance of any single result in a paper such as this is typically not massive but addressing questions larger than those studied here is quite difficult to fit into a single paper. In that sense, the results are of very reasonable significance.

**Weaknesses:**

Generally the paper does a good job of explaining high level meanings behind the more heavy annotation however the proof sketches are a fair bit more involved than I would expect. As is common these days, full proofs are not given in the paper itself but relegated to the appendices.

As noted above, the results are not hugely significant but I do not believe it is realistic to publish papers in this domain only if they are groundbreaking. The amount of work required for incremental progress is significant.

**Questions:**

I found the initial motivating example quite reasonable but it loosely acknowledges that any real election does not have an infinite number of voters. Theorem 2 goes on to rely on "sufficiently large n" which seems weakened given the motivation. Is that a reasonable criticism and does that weaken the impact of the result? (Admittedly, the theorem is already rather removed from having impacts on real-world elections)

**Limitations:**

The authors are clear about the limitations of the work.

---

> ### Author Rebuttal · Authors · 2023-08-03
>
> **Reviewer Comment:**
>
> > Theorem 2 goes on to rely on "sufficiently large n" which seems weakened given the motivation. Is that a reasonable criticism and does that weaken the impact of the result?
>
> **Response:**
>
> This requirement is theoretically necessary, because it is possible to define positional scoring rules that are arbitrarily close to Plurality (say giving some $\varepsilon \ll 1$ points to the second place candidate). However, we don't believe this greatly hinders the impact of the result, when viewed in conjunction with our other results. Indeed, Theorem 3 shows that Borda fails for any $n \ge 2$, and explicit computation shows that many other "reasonable" rules also fail for small $n$.

---

> > ### Comment · Reviewer_kNTn · 2023-08-17
> >
> > I thank the authors for their clear and sufficient response and leave this comment to acknowledge reading it. I will uphold my review as it is.

---

### Official Review · Reviewer_qq2e · 2023-07-06

**Soundness:** 3 good
**Presentation:** 3 good
**Contribution:** 2 fair
**Rating:** 5
**Confidence:** 4

**Summary:**

This paper presents several results related to strategy proof voting rules when the set of agents has correlated beliefs. The classic results in social choice theory assume that a manipulating agent has access to the entire set of preferences of all agents, while in the setting discussed in the paper we assume that agents have correlated beliefs and we want to find voting rules that are incentive compatible under a Bayesian model (typically called OBIC rules). The paper provides a positive result in that plurality is OBIC for a number of popular models (Mallows, PL) and show negative results for other positional scoring rules (PSRs) under Mallows and a few results on Copeland and Maximin.

**Strengths:**

+ The result that plurality is SP under a large space of correlated beliefs is interesting and a nicely proven result.

+ The writing is good and well presented, the paper sets itself well in the literature.

**Weaknesses:**

- My biggest comment here is fit for NeurIPS. This is a pretty straight social choice paper. While there are 2 references to ICML papers the bulk of the paper is a pretty straight statistical analysis/bounds paper (AI Stats?). This isn't all bad but it would be nice to include at least a bit more discussion on the relevance to the venue.

- The results are largely negative when we have n > 3 or n=3 in that most rules are no OBIC -- while I like the result for plurality this ties with my last point: what's the take away here for the ML community?


### Minor Issues:

* Maybe add the plurality result to Table 2 so it is complete for the entire set of results in the paper.

* "strategy-proof for a large class of beliefs containing the specific ones we introduce" --> this isn't clear in the abstract, please revise.

* Note that all the rules we consider are Pareto efficient. --> minor quibble but at line 81 where this is introduced and it is never defined or returned to.

**Questions:**

? What about Urn models? It seems like the conditions for Lemma 1/2 would hold for Urn type models but I'm not sure.

-------
After Rebuttal:

Thanks for the rebuttal and answering my questions -- I overall liked this paper and glad we could identify a place to strengthen the work.

**Limitations:**

Limitation discussion is not really there but at the end of the day the paper is well scoped so doesn't need to be added (though see weakness above).

---

> ### Author Rebuttal · Authors · 2023-08-03
>
> **Reviewer Comment:**
>
> > My biggest comment here is fit for NeurIPS. This is a pretty straight social choice paper. While there are 2 references to ICML papers the bulk of the paper is a pretty straight statistical analysis/bounds paper (AI Stats?). This isn't all bad but it would be nice to include at least a bit more discussion on the relevance to the venue.
>
> **Response:**
>
> Computational social choice in general has long been of interest to AI/ML researchers. In NeurIPS alone there have been many examples of this over the last decade.
>
> With respect to voting under stochastic ranking models, the topic we study, NeurIPS papers include:
> * "Random utility theory for social choice" (Azari Soufiani et al. 2012)
> * "Generalized method-of-moments for rank aggregation" (Azari Soufiani et al. 2013)
> * "Diverse randomized agents vote to win" (Jiang et al., 2015)
> * "Is approval voting optimal given approval votes?"" (Procaccia and Shah, 2015)
> * "Axioms for learning from pairwise comparisons" (Noothigattu et al., 2020)
>
> In terms of computational social choice more broadly, NeurIPS papers include:
> * Citizen's assemblies: "Neutralizing self-selection bias in sampling for sortition" (Flanigan et al., 2020), "Fair sortition made transparent" (Flanigan et al., 2021)
> * Participatory budgeting: "Proportional participatory budgeting with additive utilities" (Peters et al., 2021)
> * Models of representative democracy: "A mathematical model for optimal decisions in a representative democracy" (Magdon-Ismail and Xia, 2018)
> * Distortion of voting rules: "Efficient and thrifty voting by any means necessary" (Mandal et al., 2019)
> * Smoothed analysis: "The smoothed possibility of social choice" (Xia, 2020)
>
> With this in mind, the topic is quite relevant to at least a sizable subset of the NeurIPS community. We would also certainly include more of the aforementioned citations in a camera-ready revision.
>
> **Reviewer Comment:**
>
> > What about Urn models? It seems like the conditions for Lemma 1/2 would hold for Urn type models but I'm not sure.
>
> **Response:**
>
> We are not currently aware of urn models that generate rankings. However, we certainly believe that a large class of models that lead to some level of correlation (as preferrential-attachment models tend to in other settings) would satisfy Lemma 1.

---

> > ### Comment · Reviewer_qq2e · 2023-08-17
> > **Thanks**
> >
> > 1) So I wasn't asking for a list of COMSOC papers but again providing more of this in the context of the paper itself would greatly improve it's positioning. So thanks for the list but better to explain how these papers intersect with the topics of the current paper in the context of the conference.
> >
> > Either way, I leave this point for for the AC than myself.
> >
> > 2) All Urn models generate rankings.. e.g., https://www.docs.preflib.org/reference/instances/sampling.html -- not sure what you mean by not generate rankings...

---

> > > ### Author Response · Authors · 2023-08-18
> > > **Urn Model Result**
> > >
> > > Thank you for your attention to the rebuttal. We were not aware of these urn-based models of generating profiles and appreciate the pointer! Indeed, we would certainly believe such a model should satisfy the conditions of Lemma 1 as long as $r > 0$ (where $r$ is the number of balls of that color added after it is sampled).
> > >
> > > As a brief proof sketch, we likely need to make use of the [exchangeability property](https://en.wikipedia.org/wiki/Pólya_urn_model#Exchangeability) of these processes. Using this, we only need to show Lemma 1 holds for the last voter sampled, as the property implies the conditional distributions of other voters should be identical. For the last voter, the probability they observe a ranking $\sigma$ is proportional to $r \cdot N_\sigma + 1$ where $N_\sigma$ is the number of other voters with ranking $\sigma$. For Lemma 1,  using Bayes' rule, this should almost immediately imply that it is more likely $a$ has a higher plurality score than $b$.
> > >
> > > We would certainly be open to including this in a camera-ready revision as an additional model demonstrating our results (with details in the appendix, of course).

---

> > > > ### Comment · Reviewer_qq2e · 2023-08-21
> > > >
> > > > Ah nice, that seems reasonable. Would strengthen the results :-)

---

### Official Review · Reviewer_jEy3 · 2023-07-07

**Soundness:** 3 good
**Presentation:** 3 good
**Contribution:** 4 excellent
**Rating:** 8
**Confidence:** 4

**Summary:**

The paper considers a typical social choice problem: to design a voting rule that has desirable properties (e.g., it is onto and non-dictatorship) and it does not enable voters to misreport their vote for achieving outcomes that she prefers more. It is known that this is impossible in general, even if one allows the voter to have a prior knowledge on the preferences of other voters. This work focuses on a particular structure of this prior knowledge, namely that other voters' preferences are correlated to the manipulating voter's preferences according to classical models for the generation of preferences in social choice as Mallows model, Placket-Luce model, and Thurstone-Mosteller model.

Quite surprisingly, the paper proves that plurality dynamics is strategyproof when the agent has prior knowledge compatible with these models of preference generation, while other positional scoring (and Copeland and maxmin) rules are not, unless for a large number of voters.

**Strengths:**

The problem is a well-established problem in social choice, and the paper provides a very positive result (the existence of a very natural voting rule that is strategyproof in a very realistic setting).

Moreover, the result is also surprising (the fact that plurality enjoy the property, while other rules fail.) This result is in a way also robust. Indeed the author proves strategyproofness of plurality not only for the above cited models of preference generation, but for a superclass of them (that contains also noisy variations of above models whenever noise is small).

**Weaknesses:**

As recognized by the authors: the negative results only holds for three candidates (it is thus possible but conjectured to be improbable that for more than three candidates plurality is not the unique rule to enjoy all desired properties).

The paper only provides an analysis of specific voting rules, but not a characterization of strategyproof rules.

**Questions:**

I believe that there is an error in the last two equations on page 6 (even if they do not affect the final result). Indeed it should be 1/(|C|+1) (u(C) - |C|u(a)) >= |C|/(|C|+1) (u(c) - u(a)), for c with minimum u(c) among all c in C (similar for the last equation). Am I right?

**Limitations:**

See above

---

> ### Author Rebuttal · Authors · 2023-08-03
>
> **Reviewer Comment:**
>
> > I believe that there is an error in the last two equations on page 6 (even if they do not affect the final result). Indeed it should be $1/(|C|+1) (u(C) - |C|u(a)) \geq |C|/(|C|+1) (u(c) - u(a))$, for $c$ with minimum $u(c)$ among all $c$ in $C$ (similar for the last equation). Am I right?
>
> **Response:**
>
> We don't believe so. Recall that we defined the utility of a set of candidates to be the mean of the individual utilties, so $u(S) = \frac{1}{|S|}\sum_{c \in S} u(c)$. Therefore, $u(C \cup \{a\}) = \frac{\sum_{c \in C \cup \{a\}} u(c)}{|C| + 1} = \frac{(\sum_{c \in C} u(c)) + u(a)}{|C| + 1}  = \frac{|C|}{|C| + 1} \cdot u(C) + \frac{1}{|C| + 1}u(a).$ When we subtract $u(a)$ from this quantity, we get equality to $\frac{|C|}{|C| + 1} \cdot u(C) - \frac{|C|}{|C| + 1}u(a).$ We will certainly add more justification to this step in a camera-ready revision.

---

> > ### Comment · Reviewer_jEy3 · 2023-08-14
> >
> > Thanks.

---

### Official Review · Reviewer_qwQq · 2023-07-26

**Soundness:** 3 good
**Presentation:** 4 excellent
**Contribution:** 4 excellent
**Rating:** 7
**Confidence:** 3

**Summary:**

This paper studies the voting problem where agents' ranking preferences are correlated. Roughly speaking, voters do not know exactly each other's preferences; when a voter knows his/her own preference, (s)he can "infer" others' preferences. Strategyproofness is then defined with expected utilities. This is a practical setting where the classical Gibbard-Satterthwaite impossibility theorem does not apply. To model this, a prior distribution of voters' preferences is defined, and then a voter, after "receiving" his/her own preference, has a posterior belief over the preference distribution. The authors consider a wide range of distributions including Mallows (with both "confident" and "unconfident" variants), Placket-Luce, and Thurstone-Mosteller. The authors show that, among all the positional scoring rules (a positional scoring rule assigns a score to each ranking position, and the output depends only on the scores), the plurality voting rule is the only one that is "strategyproof" (or, OBIC as defined in the paper). Specifically, the authors show that the plurality voting rule is OBIC. On the other hand, with three alternatives, any other voting rule is not OBIC when the number of voters is sufficiently large. For Borda Count with three alternatives, it fails to be OBIC with any number of voters that is at least 2. One key observation is that any of the abovementioned distributions satisfies the so-called top-choice correlated property.

**Strengths:**

I believe the model studied in this paper is very reasonable and practical. The main results of this paper are neat and clean. The main message that the plurality voting rule stands out is also clean and promising. I think this paper provides a significant contribution to the social choice literature.

**Weaknesses:**

I find the result that the plurality rule being OBIC is not surprising. It is expected that the signal distributions have the top-choice correlated property, and it is also not very surprising that, with this property, plurality voting is OBIC. It may be more surprising to see that other voting rules fail to be OBIC.

**Questions:**

No question.

**Limitations:**

The authors have adequately addressed the limitations.

---

### Decision · Program_Chairs · 2023-09-21

**Decision:**

Accept (poster)

**Comment:**

This paper considers a voting problem, with correlated preferences. Each reviewer has a positive view of the paper. I therefore recommend acceptance.